# Oral Immunization with Yeast-Surface Display of SARS-CoV-2 Antigens in *Pichia pastoris* Induces Humoral Responses in BALB/C Mice

**DOI:** 10.3390/idr17050104

**Published:** 2025-08-27

**Authors:** Larissa Silva de Macêdo, Benigno Cristofer Flores Espinoza, Maria da Conceição Viana Invenção, Samara Sousa de Pinho, Lígia Rosa Sales Leal, Micaela Evellin dos Santos Silva, Beatriz Mendonça Alves Bandeira, Pedro Vinícius Silva Novis, Tiago Henrique dos Santos Souza, Julliano Matheus de Lima Maux, Jacinto da Costa Silva Neto, Antonio Carlos de Freitas, Anna Jéssica Duarte Silva

**Affiliations:** 1Laboratory of Molecular Studies and Experimental Therapy-LEMTE, Department of Genetics, Federal University of Pernambuco, Recife 50670-901, Brazil; larissa.smacedo@ufpe.br (L.S.d.M.); benigno.cristofer@ufpe.br (B.C.F.E.); maria.conceicao@ufpe.br (M.d.C.V.I.); samara.pinho@ufpe.br (S.S.d.P.); ligia.leal@ufpe.br (L.R.S.L.); micaela.evellin@ufpe.br (M.E.d.S.S.); beatriz.mendonca@ufpe.br (B.M.A.B.); pedro.novis@ufpe.br (P.V.S.N.); 2Department of Biophysics and Radiobiology, Federal University of Pernambuco, Recife 50670-901, Brazil; tiago.henriques@ufpe.br; 3Laboratory of Cytological and Molecular Research, Department of Histology and Embryology, Federal University of Pernambuco, Recife 50670-901, Brazil; julliano.maux@ufpe.br (J.M.d.L.M.); jacinto.costa@ufpe.br (J.d.C.S.N.)

**Keywords:** viral infection control, whole yeast vaccine, delivery system, spike protein, nucleocapsid protein

## Abstract

**Background/Objectives**: The pandemic caused by SARS-CoV-2 boosted the development of different vaccine models. In parallel, yeasts stand out as a vaccine platform in healthcare biotechnology. Species such as *Saccharomyces cerevisiae* and *Pichia pastoris* can express heterologous proteins, which are capable of inducing specific antibodies and can perform as an attractive vaccine vehicle with immunomodulating properties due to their cell wall composition. Furthermore, the yeast surface display system facilitates antigen presentation to immune cells. We developed an oral vaccine based on *P. pastoris* displaying a synthetic antigen composed of *Spike* and *Nucleocapsid* epitopes. **Methods:** The vaccine was administered to BALB/c mice. Systemic immune response was measured through antibody detection in blood samples, and mucosal immunity was assessed via IgA levels in feces. Histopathological analysis of intestinal and gastric tissues was also conducted. **Results:** The yeast-based vaccine elicited a humoral immune response, reflected in the production of neutralizing antibodies and elevated levels of IgG2a and IgG2. No structural alterations or pathological changes were observed in gastrointestinal tissues. **Conclusions:** This study demonstrates the feasibility of using *P. pastoris* as an oral vaccine delivery system, supporting previous findings with other yeast species such as *Saccharomyces cerevisiae*, and highlighting its potential in developing effective mucosal vaccines.

## 1. Introduction

The use of yeasts in vaccine development has gained increasing attention over recent decades due to their unique combination of safety, ease of genetic manipulation, and ability to elicit robust immune responses [1]. These eukaryotic microorganisms have shown promise both as delivery vehicles and as platforms to produce recombinant antigens [1,2]. Moreover, yeasts are capable of withstanding the harsh conditions of the gastrointestinal tract, including pH variations and digestive enzymes, making them excellent candidates for the oral administration of vaccine antigens [3,4]. The use of yeasts may also reduce production costs and eliminate the need for chemical adjuvants, thereby enhancing vaccine accessibility, particularly in resource-limited locations [1,3,5,6].

In this context, yeasts have been extensively explored as biofactories for the production of heterologous proteins, with *Saccharomyces cerevisiae* and *Pichia pastoris* (formerly known as *Komagataella phaffii*) being the most common species. *S. cerevisiae*, traditionally employed in the food and pharmaceutical industries, holds the Generally Recognized As Safe (GRAS) status granted by the U.S. Food and Drug Administration (FDA) [1,7]. *P. pastoris*, in turn, stands out for its high efficiency in recombinant protein expression, its ease of cultivation in simple media, and its ability to perform post-translational modifications similar to those of higher eukaryotic cells [8]. Both species are considered safe, non-pathogenic, and well-established in biotechnological applications [9].

Particularly, *Pichia pastoris* has been applied as a platform for the production of recombinant vaccines, including those targeting pathogens such as Zika virus, Influenza, and Severe Acute Respiratory Syndrome Coronavirus 2 (SARS-CoV-2) [5,10,11,12,13,14]. This yeast displays a higher concentration of glucans and mannans in its cell wall compared to *S. cerevisiae*, which enhances its interaction with antigen-presenting cells and the activation of T lymphocytes [15]. The adjuvancy of *P. pastoris* can be triggered through heat treatment, which inactivates its metabolic and reproductive functions without compromising structural integrity [15].

Once introduced into the host, yeast cells can be phagocytosed by innate immune cells such as macrophages and dendritic cells (DCs), which recognize and respond to immunogenic components present in the cell wall, including mannoproteins, β-d-glucans, and chitin [2,9,13]. These components interact with pattern recognition receptors (PRRs), such as Toll-like receptors (TLR2, TLR4, and TLR6), mannose receptors (MR/CD206), and Dectin-1, which is highly expressed on DCs. These interactions promote the internalization of yeast cells by antigen-presenting cells (APCs), enabling antigen processing and presentation through the classical MHC class II pathway as well as through cross-presentation mechanisms via MHC class I [9].

In the context of oral vaccines, the use of *P. pastoris* as a delivery vehicle offers additional advantages. Its cell wall acts as a protective barrier against the harsh conditions of the gastrointestinal tract, preserving antigen integrity until it reaches the mucosa-associated lymphoid tissues [4,9,12,16]. Mucosal immunity is critical for protection against pathogens that infect through mucosal surfaces such as the respiratory and gastrointestinal tracts. Additionally, yeast surface display technology improves this process, allowing the presentation of recombinant proteins on the yeast cell wall via covalent anchoring systems, such as glycosylphosphatidylinositol (GPI) anchors, which bind the antigen to the β-1,6-glucan of the cell wall. This direct display facilitates recognition by components of the immune system and enhances the immunogenicity of the vaccine formulation [6,17].

Finally, multi-epitope vaccines have shown promising results in stimulating the immune system against pathogens. They are safer and can be designed with greater precision, resulting in highly conserved immunogenic epitopes [18,19]. Depending on the epitope composition, these vaccines can induce Th1- or Th2-biased immune responses. A multi-epitope protein r4R, derived from *Leptospira interrogans*, administered subcutaneously in guinea pigs, promoted a Th1 response, observed through the IgG2a/IgG1 ratio and IFN-γ production [20], whereas Gao et al. (2024) orally immunized BALB/c mice with a vaccine composed of *S. cerevisiae* carrying a multi-epitope protein from African swine fever virus, generating a Th1 and Th2 response [21].

Given the urgent need for more effective and safer vaccines against various infectious diseases, this study proposes the use of recombinant *P. pastoris* displaying a synthetic multi-epitope antigen (Scov2-Sint1) derived from SARS-CoV-2 proteins on its surface. The vaccine was administered orally to mice for immunological evaluation. Regarding safety, histopathological analyses of the stomach and intestines of vaccinated animals revealed no gastrointestinal damage. Collectively, these findings reinforce the potential of *P. pastoris* as a safe and effective platform for oral vaccines against infectious agents, capable of inducing both systemic and mucosal immunity.

## 2. Materials and Methods

### 2.1. Gene Design

In a previous study, Invenção et al. (2025) constructed a synthetic multi-epitope SARS-CoV-2 vaccine antigen (Scov2-Sint1) to produce a DNA vaccine capable of generating both antibody-mediated and cellular immune responses [22]. In summary, epitopes present in the Spike (S) and Nucleocapsid (N) proteins that bind to Major Histocompatibility Complex (MHC) class I (MHC-I) and class II (MHC-II) and B cells were selected. The criteria used to define the epitopes for T cells were the binding capacity to more than 1 molecule of MHC (allowing broad population coverage), avoiding overlapping amino acid sequences to refine epitopes that differ from each other, and the length of each peptide to be able to bind to MHC-I (8-11 aa) and MHC-II (>11 aa). Peptides of 16 aa were defined for B cells. Other aspects considered were IC50 values <50, indicating strong binding of the peptide-MHC, and antigenicity and score values.

The multi-epitope construct presents eight Spike epitopes and seven Nucleocapsid epitopes (Figure 1). The nucleotide sequence was codon-optimized for expression in human cells. Linkers were added between the epitopes to facilitate peptide processing and recognition by the immune cells. The KK linker flanks the B cell epitopes due to its potential to target cathepsin B, an enzyme responsible for antigen presentation via [23,24]. The GPGPG linker was placed among the T cell epitopes (recognition via MHC II) to stimulate responses by helper T lymphocytes [25,26,27]. The AAY linker, present among T cell epitopes (for presentation via MHC I), is a site for lysosomal and proteasomal cleavage, that generates sites suitable for epitope transport via Transporter Associated with Antigen Processing (TAP), stimulating presentation to cytotoxic T cells [27,28].

### 2.2. Obtaining Multi-Epitope Sequence

The Scov2-Sint1 (1157 bp) was chemically synthesized by GenOne Biotechnologies (Rio de Janeiro, Brazil) and formulated into the pColaDuet-1 vector. The target gene’s design included restriction sites that allowed subcloning into the yeast expression vector pPGK-Agα, suitable for protein anchoring.

### 2.3. Obtaining Expression Vectors

The pColaDuet-1-Scov2-Sint1 vector was treated with *Xho*I/*EcoR*V for subsequent subcloning into the pPGK-Agα expression vector (De Almeida et al., 2005; non-commercial vector provided by Prof. Dr. Fernando Araripe Gonçalves Torres, University of Brasília) [29]. The final construct was used to transform chemocompetent *E. coli* Top10 cells, and clone confirmation was performed by restriction analysis with *Xho*I/*Hind*III enzymes.

The pPGKΔ3-Agα vector (3 kb) was used to allow the production and anchoring of recombinant proteins in *P. pastoris*. The expression cassette (Figure 2) presents the cloned gene under the control of the 412 bp constitutive *PGK1* (3-phosphoglycerate-kinase) promoter, a variant of the original promoter (Variant 3) that belongs to the glycolytic pathway [29,30]. This vector presents the bleomycin gene derived from *Streptoalloteichus hindustanus* (*ShBle*) as a selection marker, conferring resistance to Zeocin. In addition, it presents the signal peptide α-mating factor (α-MF), derived from *Saccharomyces cerevisiae* and codon-optimized for *P. pastoris*, necessary for the secretion of proteins produced [31]. The pPGKΔ3-Agα vector presents the sequence of the α-agglutinin (Agα) anchor protein that, when fused with the C-terminal region of the encoded protein, allows the anchoring to the β-1,6 glucans present in the yeast cell wall [32]. Additionally, it has a sequence of the histidine tag (9xHIS) that enabled immunodetection by Western blot and immunofluorescence methods.

### 2.4. Microorganism Strains and Culture Conditions

In the cloning experiments, the bacterium *E. coli* Top10 [(F-mcrA Δ (mrr-hsdRMS-mcrBC) ϕ 80lacZΔM15ΔlacX74 recA1 araD139Δ (ara leu) 7697galU galK rpsL (StrR) and A1nupG)] was used as the host organism for the recombinant plasmids. The clones transformed with the pColaDuet-1-Scov2-Sint1 vector were seeded in Luria–Bertani (LB) culture medium with the addition of 100 μg/mL of ampicillin. Colonies carrying the pPGKΔ3α vector were seeded in Luria–Bertani Low Salt (LB_LS_) with 25 μg/mL of Zeocin (Invitrogen, Carlsbad, CA, USA).

For the solid medium, 2% agar was added to the medium composition. In both cases, the culture was performed at 37 °C for 16 h, under orbital shaking (150 rpm), when necessary. For recombinant expression, the yeast *P. pastoris* GS115 (Genotype: His4 and Phenotype: Mut+ and His−) was used. The culture medium for transformation plates was YPDS (yeast extract, peptone, and dextrose, supplemented with sorbitol) plus 1.5% agar, and 100 μg/mL of Zeocin. For induction assays, yeasts were grown in YPD, at 28 °C, under orbital shaking (150 rpm).

### 2.5. Obtaining Recombinant Yeast 

Approximately 10 µg of the plasmid, pPGKΔ3α-Scov2-Sint1, was linearized with the *Sac*I enzyme, precipitated (0.3 M sodium acetate/2.5× absolute Ethanol), and homogenized in 10 µL of ultrapure water to transform *P. pastoris*. Yeasts were prepared prior to electroporation, as described by Silva et al. (2021) [11]. For the procedure, 80 µL of competent cells was mixed with 10 µL of linearized DNA. Then, 320 µL of ice-cold 1M sorbitol was added to the solution, which was transferred to a cuvette (0.2 cm) and exposed to an electrical pulse of 1500 V for 5 ms using the ECM ^®^ 399 Electroporation System (BTX™, Holliston, MA, USA). Immediately after the pulse application, 1 mL of 1M sorbitol was added. The content was incubated at 28 °C for 2 h, before being seeded on YPD plates containing Zeocin. GS115 cells electroporated without DNA were used as a negative control.

### 2.6. Western Blot Analysis

To evaluate protein production, yeasts cultivated for 72 h were harvested by centrifugation (4000× *g*/10 min), and lysed with glass beads (425–600 µm; Sigma-Aldrich, St. Louis, MO, USA), to obtain protein extracts. The cell pellets were homogenized in lysis buffer (Triton x-100 2%, SDS 1%, NaCl 100 mM, Tris-HCL pH 8 10 mM, EDTA 1 mM, 2-mercaptoethanol 10) in a proportion of 100 µL of buffer for each 1 mL of centrifuged culture, and microspheres (*v*:*v*). Then, we performed 5 cycles of 45 s of vortex agitation intercalated with ice incubation. At the end of the process, the lysates were centrifuged (12,000× *g*/10 min/4 °C), and the supernatant (protein extract) was transferred to a new tube. The protein extract was homogenized with 2× Laemmli denaturing buffer (L2x) in a 1:1 volume ratio, heated to 75 °C for 10 min, and then applied to the SDS-PAGE gel (12.5%), under 35 mA for 90 min. Then, the proteins were transferred to a polyvinylidene fluoride (PVDF) membrane under 25 volts (constant) for 35 min in a semi-dry transfer system. Immunodetection was performed after blocking the membrane with 5% milk (no fat) solution, followed by incubation with a primary monoclonal anti-histidine antibody (Sigma Aldrich, St. Louis, MO, USA), diluted 1:3000 in TBS-Tween (1% Tween diluted in PBS), for 1 h. Three washes were performed with TBS-Tween (1%) among incubations. Then, the secondary antibody Anti-Mouse IgG (Whole Molecule)—Peroxidase (Sigma Aldrich), diluted 1:5000 in TBS-Tween, was added and incubated for 1 h, with subsequent washing with TBS-Tween (1%). The signal was detected by chemiluminescence, using the Super Signal TM West Femto Maximum Sensitivity Substrate (Thermo Fisher Scientific, Waltham, MA, USA) and C-DiGit^®^Blot Scanner equipment (LICORbio, St. Lincoln, NE, USA) to digitize the images. The protein marker PageRuler Prestained Protein Ladder (Thermo Fisher Scientific, Waltham, MA, USA) was used as a molecular weight reference.

### 2.7. Immunofluorescence Analysis

Immunofluorescence microscopy was employed to confirm the exposure of the recombinant antigens on the cell wall, as described by Wasilenko et al. (2010), with some modifications [16]. After 72 h of cultivation, the cells were centrifuged at 4000× *g* for 10 min, washed in PBS 1×, and adjusted to an O.D.600 = 1. The cells were heated at 60 °C for 1 h for inactivation. The anti-histidine antibody (Sigma-Aldrich, St. Louis, MO, USA) was diluted in PBS (1:200) and incubated for 1 h under agitation. Yeasts were then washed with PBS 1× before being incubated with anti-IgG antibody conjugated with fluorescein isocyanate (FITC) (Sigma-Aldrich, St. Louis, MO, USA) diluted in PBS 1× (1:500) for 45 min, in the dark under shaking. Following these steps, the cells were washed once again (three times) with PBS 1×. Then, 10 µL of the suspension was applied to slides and coverslips, and visualized using a fluorescence microscope (Motic AE31E, Motic, Universal City, TX, USA). Images were acquired with a Moticam S6 camera using Motic Images Plus 3.0 software (Motic, Universal City, TX, USA). Non-recombinant *P. pastoris* GS115 was used as a negative control and was grown under the same conditions as the recombinant yeasts.

### 2.8. Preparation of Vaccine Doses, Oral Immunization Regimen, and Sample Collection

Initially, yeasts were inoculated in YPD medium at 28 °C under agitation at 150 rpm. After 72 h of cultivation, the cells were centrifuged at 4000× *g* for 10 min, followed by three washes using 10 mL of PBS. The cells were homogenized with sterile PBS and adjusted to reach a final concentration of 2.5 × 10^7^ cells in 0.25 mL. For metabolic inactivation of the yeasts, the cells were incubated at 60 °C for 1 h and stored at 4 °C until use (Figure 3).

For the immunization trials, 8 mice were divided into two groups (4 animals per group), all female BALB/c mice, aged 6 to 8 weeks, with use approved by the UFPE Animal Use Ethics Committee (CEUA UFPE 0054/2021). The animals were raised and kept in the bioterium of the Keizo Asami Institute (iLika, UFPE-PE, Brazil) under sterile, pathogen-free conditions. Before each vaccine administration, the mice were fasted for 3 h (water was provided ad libitum). The experimental groups were non-recombinant *P. pastoris* GS115-control (G1) and *P. pastoris* Scov2-Sint1 (G2). The vaccination regimen (Figure 4) consisted of two oral doses of 250 µL, with a seven-day interval between the doses. Twenty-one days after the second dose, all animals were anesthetized (xylazine hydrochloride (10 mg/Kg) and ketamine (115 mg/Kg)), the blood was collected by cardiac puncture, and then they were euthanized. The intestine and stomach were also removed for histopathological analysis.

### 2.9. Vaccine Safety Assessment

#### 2.9.1. Myelotoxicity Tests

The blood of all experimental animals was collected by cardiac puncture, stored in EDTA tubes, and used for the preparation of blood smears for hematological evaluation through the differential count of the slides stained based on the rapid Panoptic method [33].

#### 2.9.2. Histopathological Analyses

For histopathological analyses, samples were collected from the intestine and stomach, after prior anesthesia and subsequent euthanasia. Tissue fragments were fixed in 10% formalin for 24 h. After this process, the samples were washed with water and immersed in 70% ethyl alcohol for 3 to 4 days. Subsequently, the material was dehydrated, replacing the water present in the tissue with a dehydrating agent (ethanol), with a gradual increase in concentration (70%, 80%, 90%, 100%). The samples were clarified in xylene and subsequently embedded in paraffin. Paraffin blocks were cut using a manual microtome (Leica, Wetzlar, Germany) at a thickness of 5 µm, stained with Hematoxylin and Eosin (HE), and finally fixed with synthetic resin coverslips (Entellan/Merck, Darmstadt, Alemanha). Histopathological analysis was performed using a microscopy system with the software 2.3 ZEN (Zeiss, White Plains, NY, USA) blue-edition image capture software (©Carl Zeiss Microscopy GmbH), image capture system, and the Axio Zeiss microscope^®^ (Zeiss, Oberkochen, Baden-Württemberg, Alemanha).

### 2.10. Analysis of Fecal Samples

The procedure for the detection of immunoglobulins extracted from feces of immunized animals was based on Wang et al. (2016) [34]. Approximately 0.1 g of intestinal content was collected from mice after the first and second doses of the vaccination schedule. The extraction buffer composed of 0.01 M PBS (pH 7.4 and 0.5% Tween 20) was added to each tube (1 mL of buffer to 0.1 g of sample; *v*/*w*) and the samples were thoroughly homogenized by manual shaking and using a vortex mixer. Then, the suspensions were centrifuged at 10,000× *g* for 10 min at 4 °C. The supernatant was transferred to a new tube and stored at −80 °C for later analysis of immunoglobulin detection using the Mouse Immunoglobulin Isotyping Kit (BD^TM^ Bioscience, Franklin Lakes, NJ, USA).

### 2.11. Evaluation of the Humoral Response Induced by Vaccines

The production and activity of antibodies induced by the vaccination were evaluated from serum samples diluted in PBS 1× in a ratio of 1:10. To check the neutralizing potential of antibodies, the cPass Neutralization Antibody Detection kit (Genscript, Nanjing, China) was used, according to the manufacturer’s instructions. The kit allows measurement of the percentage of inhibition of the binding of the receptor binding domain (RBD) protein to the ACE-2 receptor in the presence of neutralizing antibodies. This RBD protein is specific for SARS-CoV-2, allowing the correlation of the immune response with the presence of neutralizing antibodies directed at the virus. Thus, the immunoglobulins detected are specific for SARS-CoV-2. The values are read by absorbance and reflect the antibody levels that bind to the RBD protein conjugated to peroxidase, and they prevented its binding to the ACE2 protein adsorbed in a well of the 96-well plate. The calculation of the neutralization index followed the formula N(%) = (1 − OD of the sample/OD of the negative control) × 100, with a 20% cutoff. The absorbance was analyzed with a microplate reader. The immunoglobulins IgG, IgM, IgA, and IgE were also measured from the serum and feces using the Mouse Immunoglobulin Isotyping Kit (BD^TM^ Bioscience), and the corresponding data were acquired by flow cytometry BD ACCURI C6 (BD^TM^ Bioscience, Franklin Lakes, NJ, USA).

### 2.12. Statistical Analysis

Graphs and statistical analyses were generated by GraphPad Prism version 9.0.0.121 (GraphPad, Boston, MA, USA). Student’s *t*-test was applied to assess statistical differences between groups. Results with a *p*-value <0.05 were considered statistically significant. To ensure the validity of Student’s *t*-test used in the comparisons, data normality analysis was performed using the Anderson–Darling, Shapiro–Wilk, Kolmogorov–Smirnov, and D’Agostino–Pearson tests.

## 3. Results

### 3.1. Confirmation of the Obtaining Expression Vectors

The plasmids fused with the Scov2-Sint1 gene had an approximate total size of 4816 bp. Cloning was confirmed by enzyme restriction analysis with the release of fragments corresponding to the sequences of the synthetic gene and the empty vector (Figure 5).

### 3.2. Recombinant Protein Production

The confirmation of protein production by recombinant yeasts was performed by Western blot, where we observed a band of approximately 60 kDa corresponding to Scov2-Sint1 (Figure 6). The non-recombinant *P. pastoris* GS115 was used as a negative control.

### 3.3. Immunofluorescence

Fluorescence microscopy was used to identify the anchoring of recombinant protein in the yeast surface (Figure 7). Fluorescence generated by antibody labeling was observed only on slides containing recombinant yeast (Scov2-Sint1). Since no fluorescence was detected on slides containing non-recombinant *P. pastoris* GS115, the results indicate that the interaction was specific for the recombinant protein displayed on the cell wall.

### 3.4. Vaccine Safety Profile

Throughout the vaccination regimen, the animals’ weight was assessed (data presented in Appendix A), leading to the observation of an increase in all groups, following the same pattern and showing linearity with the animals’ lifespan. No other events, such as behavioral changes or physical alterations, nor signs of inflammation or swelling, were identified in the animals.

In the hematological analysis, the immunized mice did not show relevant changes between the groups regarding differential leukocyte count. Furthermore, leukocyte values (Appendix A) were within the reference value [35,36].

According to the histopathological analysis (Figure 8), the administration of recombinant yeast-based vaccines did not provoke damage to the stomach and intestine, which suggests the safety of these vaccine preparations. Thus, for all groups, no structural changes or reactive (infiltrated) or degenerative processes were observed, indicating the preservation of the mucosa, submucosa, and adventitia layers.

### 3.5. Assessment of the Humoral Immune Response Induced by Vaccines

To evaluate the humoral response triggered by the vaccine candidate, blood samples from immunized animals were collected at the endpoint of the vaccination schedule and analyzed for antibody production. Mice from the experimental group presented higher percentages of inhibition of the binding of the RBD protein to the ACE-2 receptor, suggesting the presence of neutralizing antibodies (Figure 9), despite the absence of statistical significance between the groups.

*P. pastoris*:Scov2-Sint1 induced the production of some immunoglobulin isotypes, such as IgG2a (Figure 10B) and IgG2b (Figure 10C). There was no significant release of IgA (Figure 10E), although there is an indication of higher production induced by the synthetic antigen. The calculation of the IgG2a/IgG1 ratio (Figure 10H) helps evaluate trends in Th1 (>1) or Th2 (<1) responses, and the vaccine with the anchored synthetic antigen demonstrated a biased Th1 profile that must be further investigated through assays that evaluate the cellular immune response and cytokine profile. The analyses regarding IgG3, IgM, and IgE (Figure 10D,F,G) showed no significant differences between the groups.

In fecal samples (Figure 11), IgA levels in mice immunized with Scov2-Sint1 were significantly higher than in mice treated with the control group. In comparison, there was an increase in IgA after the second dose of the vaccine, indicating the importance of the booster dose. No significant changes were observed in the levels of IgM, IgE, and IgG isotypes (data presented in Appendix A).

## 4. Discussion

In our study model, recombinant *P. pastoris* GS115 was employed to produce and display on the surface a synthetic multi-epitope antigen (Scov2-Sint1), which successfully induced both humoral and mucosal immune responses. These findings reinforce the efficacy of the yeast-based platform as a promising system for oral antigen delivery and extend the observations of our previous study [22], in which the same antigen, administered intramuscularly as a DNA vaccine, elicited effective T cell-mediated responses involving both CD4^+^ and CD8^+^ lymphocytes [22].

Oral administration represents a particularly attractive strategy for inducing mucosal immunity, a critical aspect of protection against respiratory pathogens, such as SARS-CoV-2, which primarily infect the host through mucosal surfaces. Our results are consistent with those of Chen et al. (2025), who demonstrated the potential of antigen-anchored yeasts to elicit strong intestinal mucosal immunity [12]. Similarly, our vaccine induced elevated levels of fecal IgA, suggesting effective activation of the local immune response [12]. While Chen et al. (2025) highlighted the superiority of anchored antigen presentation over secreted forms and even over other platforms (such as nanoparticles and bacterial vectors), our study also reinforces this concept by demonstrating that *P. pastoris* can induce mucosal immunity [12].

Furthermore, our findings align with those of Zhang et al. (2022), who have shown that recombinant yeasts elicit specific IgG and IgA responses against the SARS-CoV-2 RBD, positively modulating the gut microbiome [37]. Although our study did not include microbiota analyses, the absence of histological adverse effects and the preservation of tissue integrity suggest good tolerability of the *P. pastoris*-based oral vaccine, paving the way for future investigations into its impact on the gastrointestinal microenvironment.

Regarding protein production, detection occurred in yeast cells cultured for 72 h at 28 °C, optimal conditions for protein production in *P. pastoris* [38,39,40]. It is worth highlighting the ability of *P. pastoris* to generate proteins similar to those commonly produced in cells of higher eukaryotes, both in function and structure [41,42,43]. Anchoring proteins on the yeast surface represent an efficient system as a vaccine model, as it improves the interaction between the antigen and dendritic cells [44,45].

When tested against H5N1 and H7N9 Influenza viruses, this strategy demonstrated immunological protection after vaccination and infectious challenge [16,46,47]. Furthermore, *S. cerevisiae* anchoring the RBD region, wild-type, and variant sequences (B.1.1.7, B.1.351, and B.1.671.1), administered subcutaneously, induced activation of humoral and cellular responses with the production of neutralizing antibodies against SARS-CoV-2 [48].

In this system, the surface-displayed antigen maintains its antigenic capacity after passing through the stomach and can remain in the intestinal tract for five days after vaccination. Indeed, this presentation can favor the recognition by antigen-presenting cells (APCs) [4,45,49,50,51]. It is worth noting that Zhang et al. (2022) demonstrated that yeast administration can aid in the reconstruction of the microbiota, besides immunological protection, in a murine model [37].

Oral vaccines can activate the immune system against the target pathogen. In the gut-associated immune tissues (GALTs), the germinal centers in Peyer’s patches facilitate the triggering of humoral and cellular responses [52,53]. In the last ten years, several studies involving oral vaccines using yeast as a delivery system have been published, as reviewed by Austriaco (2023) [4]. Yeast cells have an adequate size for internalization by M cells and, once used as vaccine vehicles, can induce the production of specific IgA and IgG and lead to the development of local and systemic neutralizing immune responses [54,55].

The immunoglobulin isotypes detected after vaccination reinforce that fact that our vaccine model can stimulate the humoral response in mice. The difference in the presence of IgA in the serum and the feces (Figure 10E; Figure 11) may have occurred due to the normal physiological process in the production of immunoglobulins and their function in the tissues analyzed. Seric IgA appears much more quickly after infection, but its concentration decreases rapidly [56,57,58].

The second immunoglobulin to show up in the serum after IgA in the blood is IgG, which lasts for 200 days following SARS-CoV-2 virus infection [59,60]. Vaccination carried out by the intramuscular or oral route enables the immune system to generate specific IgG against the SARS-CoV-2 antigen and its presence both in the blood and mucous membranes of the vaccinated individual [61,62,63].

In severe infection processes caused by SARS-CoV-2, there is a decrease in the presence of Th1 lymphocytes due to a deficit in the interaction of DCs with IgG1 through its FcyR receptor [64]. Interestingly, here, when the IgG2a/IgG1 ratio was evaluated, the vaccine showed a tendency towards the Th1 profile (Figure 10H) probably associated with the presence of Nucleocapsid epitopes in the Scov2-Sint1 sequence, although this observation should be confirmed by assessing the cellular immune response. This antigenic structure promotes and stimulates a TCD8+ cellular response, unlike the Spike protein, whose structure directs the activation of TCD4+ [65,66].

Finally, the IgG subclasses with the higher effector capacity, crucial for maintaining immune protection, are IgG2a and IgG2b that interact with activating receptors (FcγRI, FcγRIII, and FcγRIV) of immune cells [67]. Our experimental vaccine induced a significantly higher increase in the levels of IgG2a and IgG2b compared to the control group (Figure 10B,C). Similar results were reported in a study that evaluated intramuscular and intranasal vaccination with the SARS-CoV-2 RBD antigen in female BALB/c mice [68,69]. Overall, we demonstrated that the recombinant *P. pastoris* Scov2-Sint1 vaccine has significant immunogenic properties, inducing the production of IgG and IgA antibodies with mucosal and systemic responses.

The induction of mucosal immune responses, especially those mediated by secretory IgA, represents one of the main objectives of oral vaccination strategies, since this immunoglobulin can neutralize pathogens directly on epithelial surfaces, preventing their adhesion and invasion without promoting local inflammation [52,70,71]. In our experiments, IgA analysis in fecal samples was used as an indicator of intestinal immune activation, considering the relevance of the mucosa–gut axis and the central role of IgA in protecting against infections at mucosal surfaces. The results revealed increased IgA levels in the group immunized with the recombinant yeast *P. pastoris*, suggesting the induction of a mucosal immune response.

Similarly, Wang et al. (2016) described the development of an oral vaccine based on *Pichia pastoris* GS115 expressing the S1 gene of porcine epidemic diarrhea virus (PEDV), which induced high levels of IgA in immunized mice and piglets [34]. In our study, IgA was detected 21 days after the first dose, with significantly higher levels compared to the seventh day after the beginning of the vaccination schedule. Despite the lack of an evaluation of the kinetics of IgA production, the data reinforce the potential of the yeast platform as an effective immunizing vehicle for inducing mucosal immunity.

In the study conducted by Gao et al. (2021), a vaccine based on *Saccharomyces cerevisiae* (EBY100/pYD1-RBD) was administered orally to mice, and immune responses were assessed by detecting IgA in fecal samples [62]. The results showed that the oral administration of recombinant yeast induced a humoral and mucosal immune response, with a significant increase in secretory IgA levels, particularly after the booster dose, a pattern similar to that observed in our study. These findings reinforce the importance of the prime-boost regimen in inducing mucosal immunity, especially considering the tolerogenic nature of the gastrointestinal tract.

Despite the promising results, it is important to highlight that these studies share methodological limitations. Among them are the lack of experiments presenting more in-depth analyses of the relationship between the systemic (IgG) and mucosal (IgA) responses, whose coordinated role remains poorly understood, and the understanding of the optimal time between doses by evaluating different vaccination schedules. Our findings corroborate the results in the literature and reinforce the fact that recombinant yeast-based strategies represent a promising alternative for inducing local immune responses with potential for oral vaccines aimed at preventing infections by pathogens that affect mucosal surfaces.

Regarding safety, animals immunized with yeast-based vaccine preparations did not present adverse changes concerning clinical signs (weight, behavior, and physical aspects) and hematological and histopathological signs (intestine and stomach), which corroborates studies that describe the non-pathogenicity of *P. pastoris* and its GRAS status. Becerril-García and collaborators (2022) evaluated the intravenous (i.v.) administration of live *P. pastoris* cells to track dissemination in female BALB/c mice and to assess the immune response raised against the yeast [72]. In this study, it was observed that the yeast is capable of disseminating to the heart, kidney, spleen, and liver, but is rapidly eliminated in the first 48 h of infection, highlighting the absence of tissue lesions in the evaluated organs [72]. This suggests that the yeast does not form permanent colonies in muscles or other organ tissues. Furthermore, these findings corroborate the GRAS status of *P. pastoris*, indicating its safety for use, including in vaccine contexts, as well as its versatility in being explored through different delivery routes, such as oral, intramuscular, and intravenous [34,72,73,74].

Although these preclinical results demonstrate the safety of the formulation and its ability to induce both mucosal and systemic immune responses, we recognize that several steps are still necessary to advance toward clinical application. Among the main limitations is the need for in vivo challenge studies to assess protective efficacy against SARS-CoV-2 infection, an approach considered the gold standard in vaccine validation. Although our work does not include detailed cellular profile analyses or antigen-presenting cell activation assays, we recognize the importance of elucidating these mechanisms, which are fundamental for viral clearance and long-term immune protection. As part of our ongoing studies, we are conducting in vitro and in vivo experiments to investigate the uptake of *Pichia pastoris* by dendritic cells and macrophages, as well as the associated activation markers and antigen presentation pathways involved. Additionally, we emphasize the importance of assessing antigen cross-reactivity with emerging SARS-CoV-2 variants, a crucial step in validating the breadth and robustness of the immune response generated. Another point of investigation is the understanding of how the yeast is processed after oral immunization. This is important to clarify some aspects of our study, including the high variability of neutralization levels among the mice that received the vaccine. Some methods for evaluating the distribution of yeast-based vaccines in the gastrointestinal tract may be feasible, as observed in the study by Chen et al. (2025), that evaluate the administration of recombinant yeasts labeled with the fluorophore DiD (1,1′-dioctadecyl-3,3,3′,3′-tetramethylindodicarbocyanine, 4-chlorobenzenesulfonate salt) [12]. Mice received a single oral dose of these labeled strains, and samples of Peyer’s patches (PPs), mesenteric lymph nodes (MLNs), and intestine were collected after 6, 12, and 24 h for analysis of the distribution in the gastrointestinal tract. In a second experiment, designed to evaluate cellular internalization, mice received YVm orally, and after 9 h, the gastrointestinal tract and PPs were collected, sectioned, and analyzed by confocal microscopy for fluorescence quantification [12].

We believe our results contribute to the advancement of biotechnology-based vaccine platforms and provide a basis for future translational research aimed at developing affordable, safe, and effective oral vaccines against SARS-CoV-2 and, potentially, other mucosal pathogens.

## 5. Conclusions

The vaccine antigen Scov2-Sint1, using the yeast *P. pastoris* as a vehicle in a surface display system, was shown to be viable and safe during the period of the proposed vaccination regimen. The activation of humoral immunity induced by the recombinant vaccine was similar to responses observed with different vaccines produced against SARS-CoV-2. The results presented here suggest that the proposed vaccine induces immune responses that must be evaluated to determine their efficacy in protecting against SARS-CoV-2 infection after a new vaccine schedule with viral challenge and cellular response evaluations.

Our study confirms that the antigen produced in *P. pastoris* is effective in inducing IgG and IgA antibodies, both in the mucous membranes and in the blood. Most studies on oral vaccines based on *P. pastoris* demonstrate the use of this species for the production of antigens used as subunit vaccines. Here, we also demonstrate the applicability of this yeast as a whole-cell vaccine administered orally, highlighting its potential for new vaccine strategies against several infectious diseases.

Importantly, this strategy is particularly relevant for pathogens that have tropism to mucosal surfaces, such as the respiratory, gastrointestinal, and genitourinary tracts. Diseases such as Coronavirus Disease 2019 (COVID-19), Influenza, and Human Papillomavirus (HPV)-associated infections could greatly benefit from mucosal immunization, which promotes local immune responses capable of preventing pathogen entry and colonization at the initial site of infection. Therefore, oral vaccination using *P. pastoris* as a delivery system may offer an effective, non-invasive, and broadly applicable platform for the development of mucosal vaccines.

## Figures and Tables

**Figure 1 idr-17-00104-f001:**
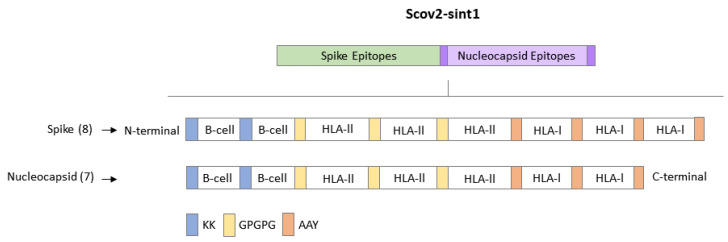
Distribution of epitopes used to assemble the Scov2-Sint1 gene.

**Figure 2 idr-17-00104-f002:**
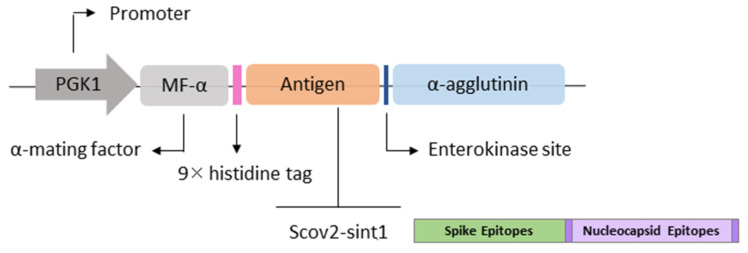
The distribution of components in the expression vector sequence arrangement. PGK1: strong constitutive promoter; MF-α: signal peptide for protein secretion; 9× histidine tag: histidine tag that allows immunodetection of the recombinant protein; Antigen: target gene (Scov2-Sint1); Enterokinase (EK) site: allows cleavage between the vaccine antigen and the anchor protein; α-agglutinin (Agα): anchor protein.

**Figure 3 idr-17-00104-f003:**
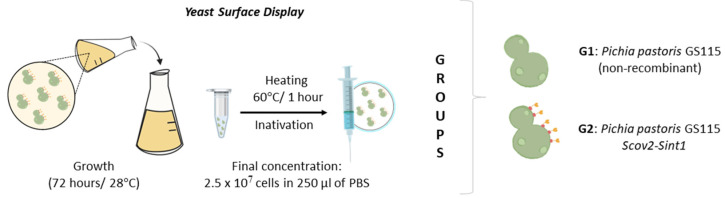
Preparation, dose adjustment, and determination of vaccine groups.

**Figure 4 idr-17-00104-f004:**
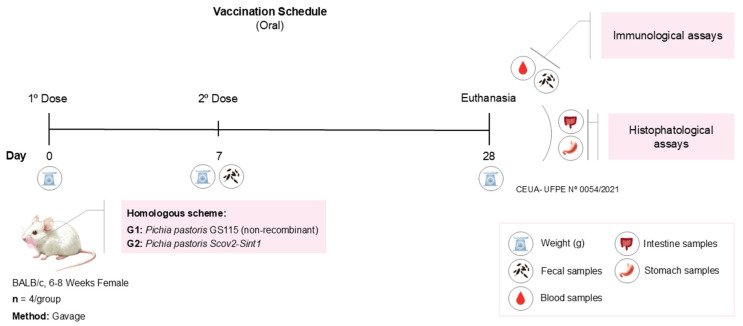
Oral vaccination schedule applied in a murine model against SARS-CoV-2.

**Figure 5 idr-17-00104-f005:**
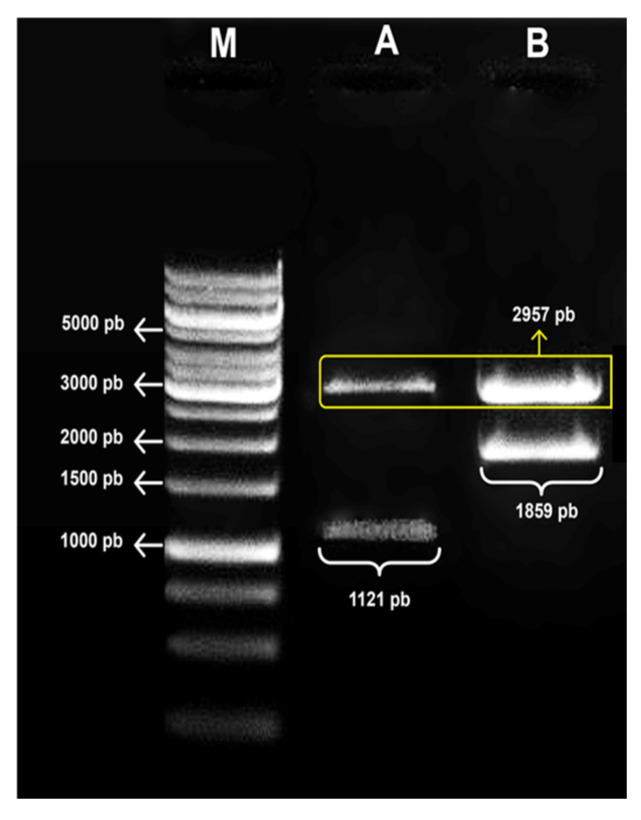
Confirmation of expression cassettes. Enzymatic treatments with *Xho*I and *Hind*III for confirmation of bacterial clones with the vectors pPGK∆3Agα (control) and pPGK∆3α-Scov2-Sint1. (M) Marker: 1 Kb (Ladder Gene Ruler, Thermo Scientific); 1% agarose gel. (A) pPGK∆3Agα (empty vector; 2957 bp), releasing the Agα region (1121 bp); (B) digestion of pPGK∆3-Scov2-Sint1 (4816 bp), releasing Agα + Scov2-Sint1 (1859 bp) and the empty vector (2957 bp).

**Figure 6 idr-17-00104-f006:**
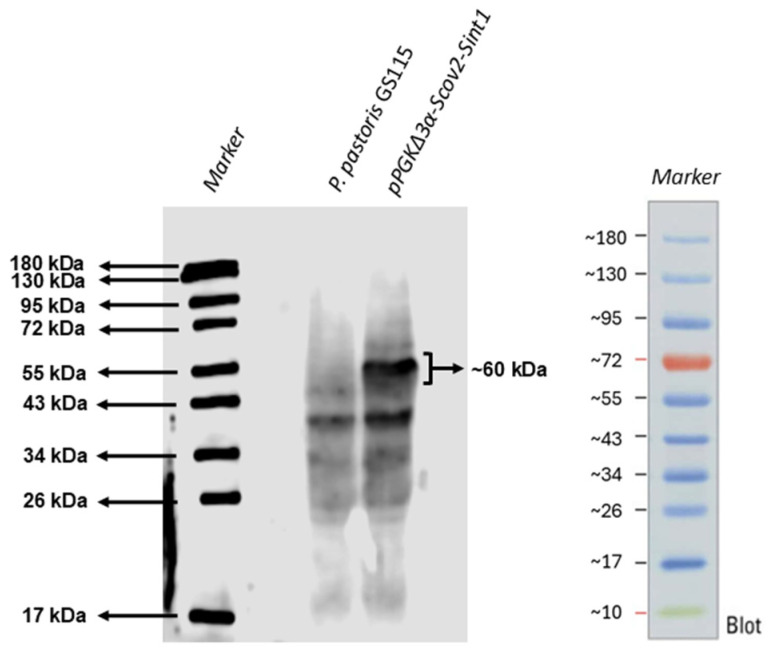
Detection of Scov2-Sint1 protein expression in recombinant *P. pastoris* by Western blot. After protein extraction and Western Blot analysis, it was possible to observe a band of approximately 60 kDa corresponding to Scov2-Sint1 + Agα labeled with anti-histidine (primary antibody), directed to HIS-tag, and anti-IgG conjugated to horseradish peroxidase (HRP). Marker: PageRuler™ Plus Prestained Protein Ladder (Thermo Scientific, Waltham, MA, USA). Control: non-recombinant *P. pastoris* GS115.

**Figure 7 idr-17-00104-f007:**
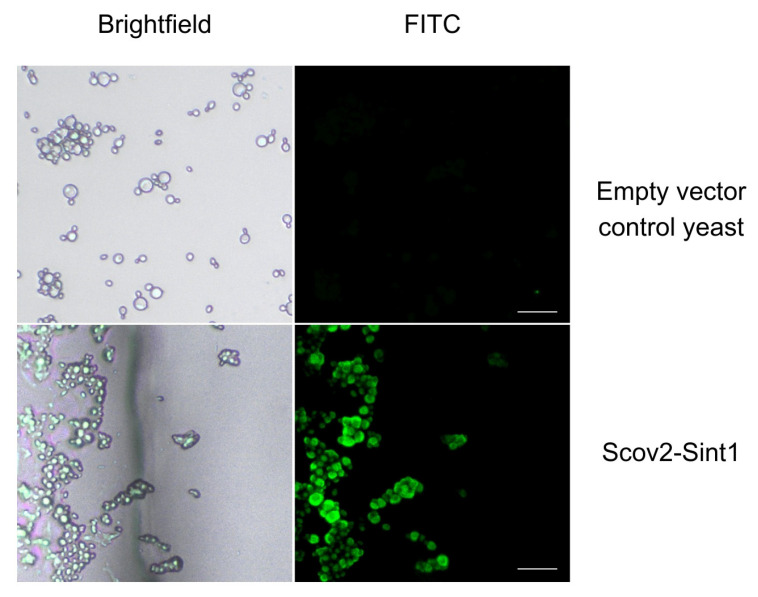
Immunofluorescence microscopy of recombinant yeast. Observation of the fluorescence on yeasts labeled with anti-HIS monoclonal antibody followed by incubation with an FITC-conjugated secondary antibody (anti-IgG), indicating the anchoring of target proteins in the yeast cell wall. Non-recombinant *P. pastoris* was used as a negative control (empty vector control yeast). Images were captured through a fluorescence microscope (Motic AE31E), at 40× magnification, with the Moticam S6 camera using Motic Images Plus 3.0 software. Scale bar: 10 μm.

**Figure 8 idr-17-00104-f008:**
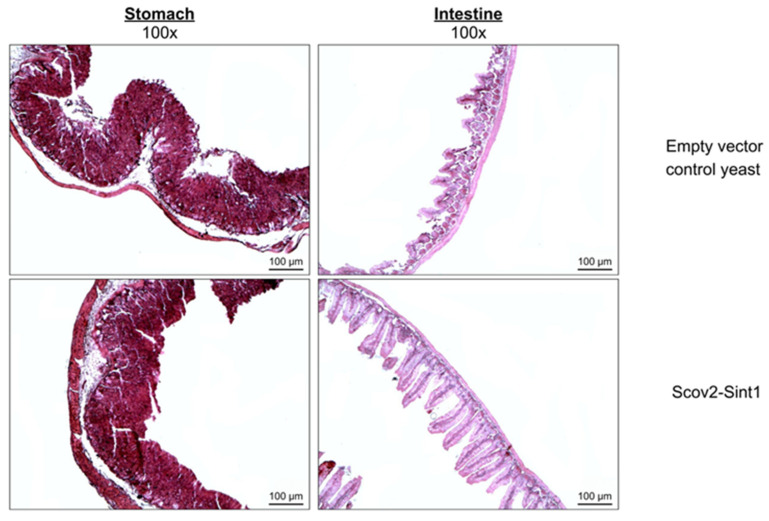
Representative histopathological analysis of the stomach and intestine of immunized mice. The following groups are represented: upper images—non-recombinant *P. pastoris* GS115 (empty vector control yeast); lower images—*P. pastoris* expressing Scov2-Sint1. Structures are shown at 100× magnification from Hematoxylin and Eosin-stained slides observed by light microscopy. Microscopy was performed with the ZEN blue-edition image capture software (©Carl Zeiss Microscopy GmbH), Zeiss^®^ Axio microscope, and image capture system.

**Figure 9 idr-17-00104-f009:**
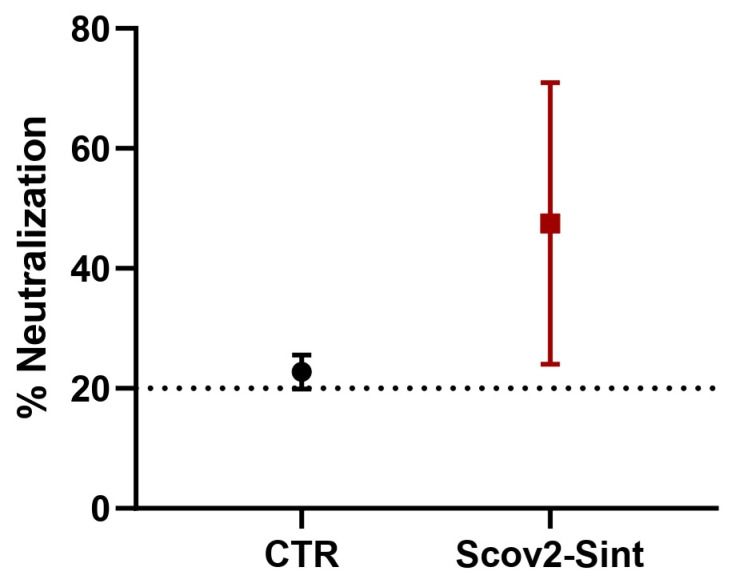
Neutralization potential. The *y*-axis corresponds to the percentage of neutralization of antibodies produced after immunization. The dotted line represents the cutoff value for detecting SARS-CoV-2 neutralizing antibodies, set at 20% signal inhibition. CTR: empty vector control yeast.

**Figure 10 idr-17-00104-f010:**
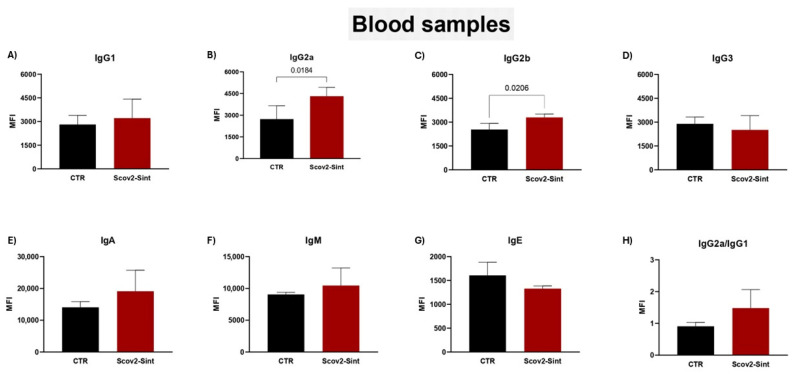
Analysis of immunoglobulin isotypes from animal serum collected 21 days after the application of the second vaccine dose. (**A**) IgG1; (**B**) IgG2a; (**C**) IgG2b; (**D**) IgG3; (**E**) IgA; (**F**) IgM; (**G**) IgE; and (**H**) IgG2a/IgG1 ratio. CTR: empty vector control yeast. MFI corresponds to the mean fluorescence intensity. Statistical significance was determined by Student’s *t*-test, considering *p* < 0.05. Bars indicate the mean value ± standard deviation.

**Figure 11 idr-17-00104-f011:**
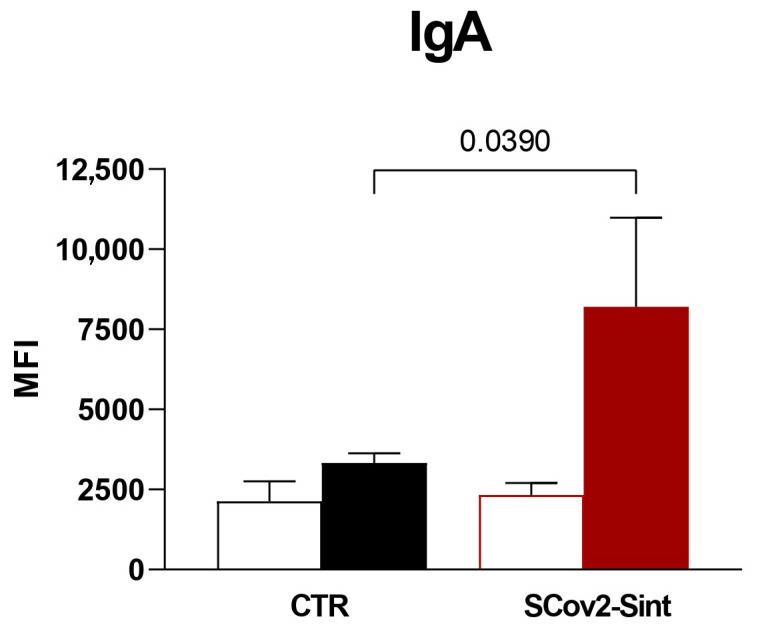
Evaluation of IgA detected in the feces of animals 7 days after the first dose (white columns) and 21 days after the second dose (filled columns). CTR: empty vector control yeast. MFI corresponds to the mean fluorescence intensity. Statistical significance was determined by Student’s *t*-test, considering *p* < 0.05. Bars indicate the mean value ± standard deviation.

## Data Availability

All relevant data from this study are available from the corresponding author.

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
