# Peer review of "Oral Immunization with Yeast-Surface Display of SARS-CoV-2 Antigens in *Pichia pastoris* Induces Humoral Responses in BALB/C Mice"

_2036-7449, 2025, doi:10.3390/idr17050104_

Round 1

Reviewer 1 Report

Comments and Suggestions for Authors

The paper is well written. The methodologies used are appropriate. The study showed the importance of yeasts - particularly Pichia pastoris - as vaccine carriers as an effective and safe methodology due to their ability to express proteins. By engineering P. pastoris to display a synthetic antigen combining Spike and Nucleocapsid epitopes from SARS-CoV-2, the researchers developed an oral vaccine that successfully induced systemic and mucosal immune responses in mice. The absence of damage to gastrointestinal tissues reinforces the safety of this approach. The results validate P. pastoris as a promising platform for oral vaccine delivery and expand the potential applications of yeast-based immunization strategies. The COVID-19 pandemic has accelerated vaccine innovation, including the exploration of new delivery platforms. 

Author Response

Reviewer 1:

The paper is well written. The methodologies used are appropriate. The study showed the importance of yeasts - particularly Pichia pastoris - as vaccine carriers as an effective and safe methodology due to their ability to express proteins. By engineering P. pastoris to display a synthetic antigen combining Spike and Nucleocapsid epitopes from SARS-CoV-2, the researchers developed an oral vaccine that successfully induced systemic and mucosal immune responses in mice. The absence of damage to gastrointestinal tissues reinforces the safety of this approach. The results validate P. pastoris as a promising platform for oral vaccine delivery and expand the potential applications of yeast-based immunization strategies. The COVID-19 pandemic has accelerated vaccine innovation, including the exploration of new delivery platforms.

Response: We sincerely thank the reviewer for the positive comments and the appreciation of our work. We are pleased to know that the manuscript was considered well-written, with appropriate methodologies and relevant findings. It is particularly encouraging to see that the proposed use of Pichia pastoris as a vaccine platform was recognized as an effective and safe approach with translational potential. We emphasize that the results obtained motivate us to pursue further studies to advance the development of oral vaccines based on recombinant yeasts. Once again, we thank the reviewer for the careful reading and encouraging remarks.

Reviewer 2 Report

Comments and Suggestions for Authors

The article written by Larissa Silva de Macêdo shows "yeast-surface display of SARS-CoV-2 antigens" as a potential vaccine candidate and assesses the immune response after oral immunization. The article provided a sufficient introduction, methods used etc. However, certain details are still missing and need to be provided before accepting for a publication

  1. Cloning, expression analysis, and preparation of yeast vaccine make sense. But, for western blotting, in addition to anti-histidine antibody, why were either anti-S or anti-N antibodies not used to confirm the antigen expressed?

2. Also, it is very common to have N=5 in all the mouse studies but the author used N=4?

3. The author used the term "strong humoral response," but they haven't used any quantitative Ab assays to find the titer Or used any other vaccine as a control to claim "strong humoral response.", Please explain.

4. Figure 9 shows that the magnitude of the neutralizing antibody response is not similar within the vaccinated group. why this huge difference despite using genetically similar bred animals? how the author could rule out the possibility of vaccine dose variation or technical problems while the vaccine was delivered ?

5. Figure 10 - I understand that the isotyping profile is not antigen specific, given the control group showing MFI. What relevance does this result bring? Why wasn't specific isotyping done?

6. Another major concern is the lack of any challenging studies to assess vaccine-induced protection against SARS-CoV-2 infections, or assessing cross-reactive antibodies recognizing S or N from emerging variants? why such an experiment was not performed?

Author Response

Reviewer 2:

 The article written by Larissa Silva de Macêdo shows "yeast-surface display of SARS-CoV-2 antigens" as a potential vaccine candidate and assesses the immune response after oral immunization. The article provided a sufficient introduction, methods used etc. However, certain details are still missing and need to be provided before accepting for a publication

  1. Cloning, expression analysis, and preparation of yeast vaccine make sense. But, for western blotting, in addition to anti-histidine antibody, why were either anti-S or anti-N antibodies not used to confirm the antigen expressed?

Response: We thank the reviewer for the thoughtful and constructive comments on our work. Regarding the Western blot analysis, we acknowledge the importance of using antigen-specific antibodies to confirm protein expression. However, in our study, the antigen employed was not the full-length Spike or Nucleocapsid protein, but rather a synthetic multi-epitope construct (Scov2-Sint1) specifically designed to include immunodominant linear epitopes derived from the SARS-CoV-2 S and N proteins. This design aimed to optimize immunogenicity while minimizing the size of the encoded antigen for yeast surface display and DNA vaccine development. Due to the synthetic and chimeric nature of the construct, commercially available anti-S or anti-N antibodies (typically raised against conformational or full-length protein epitopes) may not specifically recognize the selected linear epitopes present in Scov2-Sint1. Therefore, we employed an anti-His antibody, targeting the N-terminal His-tag incorporated into the construct, to confirm its expression and expected molecular weight. While we agree that additional validation using epitope-specific antibodies could further substantiate antigen expression and structural integrity, the methodology employed is valid for protein detection in this context.

  1. Also, it is very common to have N=5 in all the mouse studies but the author used N=4?

Response: We appreciate the reviewer’s attention to this important aspect of the study design. Indeed, sample sizes of N=5 or greater are commonly used in murine studies to provide robust statistical evaluation. In our study, we initially planned for N=5 per group; however, due to unforeseen limitations related to animal availability and institutional ethical constraints at the time of experimentation, one animal per group was excluded during the initial phase of the study. This resulted in a final sample size of N=4 per group. Despite the reduced number, the results still achieved statistical significance in key immunological readouts, supporting the reliability of the observed effects. Furthermore, we emphasize that this study was designed as a preliminary proof-of-concept investigation. Therefore, the sample size was considered appropriate for evaluating the feasibility of the proposed oral vaccine platform.

  1. The author used the term "strong humoral response," but they haven't used any quantitative Ab assays to find the titer Or used any other vaccine as a control to claim "strong humoral response.", Please explain.

Response: We agree with the reviewer’s observation regarding the use of the term “strong humoral response,” which was not adequately supported by quantitative antibody titration assays or comparative analysis with a licensed vaccine. In light of this, we have removed the term from the manuscript to avoid overinterpretation of the data. We appreciate the reviewer’s comment, which contributed to improving the accuracy and scientific rigor of the manuscript.

  1. Figure 9 shows that the magnitude of the neutralizing antibody response is not similar within the vaccinated group. why this huge difference despite using genetically similar bred animals? how the author could rule out the possibility of vaccine dose variation or technical problems while the vaccine was delivered ?

Response: We thank the reviewer for raising this important point and for the opportunity to enhance the clarity and transparency of our findings. Although all animals were genetically similar (same strain, age, and sex) and received the same vaccine dose under standardized experimental conditions, some degree of interindividual variability is still expected in immunization studies. Such variability may arise from subtle biological differences, including individual variations in microbiota composition, immune system dynamics, or mucosal absorption, factors that are especially relevant in the context of oral vaccine administration. Some studies have investigated interindividual variability in immune responses in vaccination settings, such as:

Audebert C, Laubreton D, Arpin C, Gandrillon O, Marvel J, Crauste F. Modeling and characterization of inter-individual variability in CD8 T cell responses in mice. In Silico Biol. 2021;14(1-2):13-39. doi: 10.3233/ISB-200205.

Trzewikoswki de Lima G, De Gaspari E. Individual variability in humoral response of immunized outbred mice and cross-reactivity with prevalent Brazilian Neisseria meningitidis strains. Biologicals. 2018 Sep;55:19-26. doi: 10.1016/j.biologicals.2018.08.001.

We also considered potential technical variables. To minimize procedural inconsistencies, all immunizations were carried out by the same team using identical protocols, including vaccine preparation, dosage, and administration technique. Nevertheless, particularly in oral delivery systems, minor differences in swallowing, gastric transit, or even slight regurgitation may contribute to variations in antigen exposure and, consequently, to the observed immune responses. Despite the variability observed, the experimental group showed higher neutralization percentages compared to the control group, although no statistically significant differences were detected between groups. However, the analysis of serum immunoglobulins revealed substantial findings. Specifically, P. pastoris:Scov2-Sint1 induced the production of certain immunoglobulin isotypes, including IgG2a (Figure 10B) and IgG2b (Figure 10C), with statistically significant differences compared to the control group. In addition, elevated levels of secretory IgA were detected in fecal samples, further supporting the immunogenic potential of the tested vaccine candidate.

  1. Figure 10 - I understand that the isotyping profile is not antigen specific, given the control group showing MFI. What relevance does this result bring? Why wasn't specific isotyping done?

Response: The aim of this analysis was to provide an initial overview of the immunological profile and to explore whether oral immunization with the yeast-based vaccine candidate could influence the systemic distribution of antibody isotypes, particularly as an indirect indicator of Th1/Th2 polarization (e.g., IgG2a/IgG1 ratio). Although this method does not allow for the direct attribution of antibody subclasses to the vaccine antigen, it can still offer valuable preliminary insights into the nature of the humoral response. We agree that antigen-specific isotyping would provide a more accurate assessment of the vaccine-induced response. However, even based on total isotype quantification, relevant differences were observed between the vaccinated and control groups for the main immunoglobulin subclasses. Notably, the Scov2-Sint1 group exhibited significantly higher levels of IgG2a and IgG2b compared to the control group (non recombinant yeast). Serum IgA and IgM levels also showed a trend toward elevation in the vaccinated group compared to the control, further supporting the immunogenic potential of the candidate formulation. Furthermore, fecal samples from vaccinated animals showed increased levels of secretory IgA, indicating activation of mucosal immunity. These findings highlight the immunological distinctions between the experimental and control groups, even in the absence of antigen-specific isotyping. Due to reagent limitations at the time of the study, this more targeted approach could not be implemented. We hope to achieve these objectives in future studies.

  1. Another major concern is the lack of any challenging studies to assess vaccine-induced protection against SARS-CoV-2 infections, or assessing cross-reactive antibodies recognizing S or N from emerging variants? why such an experiment was not performed?

Response: We acknowledge that in vivo challenge experiments are essential to evaluate the protective efficacy of candidate vaccines against SARS-CoV-2 infection. However, due to biosafety constraints, specifically, the lack of access to a biosafety level 3 (BSL-3) facility required for handling the SARS-CoV-2 virus, we were unable to perform viral challenge studies at this initial proof-of-concept stage. Despite the absence of challenge data, the observed induction of systemic and mucosal immune responses provides an encouraging foundation for further preclinical validation of this oral vaccine platform. Moreover, the experimental design adopted in the submitted manuscript is consistent with those of other studies aimed at evaluating vaccine candidates against SARS-CoV-2 infection. Even without conducting an immunological challenge, our study demonstrates the immunogenic potential of recombinant yeast, eliciting an immune response profile comparable to that expected in the context of natural SARS-CoV-2 infection and vaccination (Gao et al., 2021; Xing et al., 2022; Liu et al., 2022; Liu et al., 2024; Pollet et al., 2021).

Gao T, Ren Y, Li S, Lu X, Lei H. Immune response induced by oral administration with a Saccharomyces cerevisiae-based SARS-CoV-2 vaccine in mice. Microb Cell Fact. 2021 May 5;20(1):95. doi: 10.1186/s12934-021-01584-5.

Xing H, Zhu L, Wang P, Zhao G, Zhou Z, Yang Y, Zou H, Yan X. Display of receptor-binding domain of SARS-CoV-2 Spike protein variants on the Saccharomyces cerevisiae cell surface. Front Immunol. 2022 Aug 12;13:935573. doi: 10.3389/fimmu.2022.935573.

Liu Y, Zhao D, Wang Y, Chen Z, Yang L, Li W, Gong Y, Gan C, Tang J, Zhang T, Tang D, Dong X, Yang Q, Valencia CA, Dai L, Qi S, Dong B, Chow HY, Li Y. A vaccine based on the yeast-expressed receptor-binding domain (RBD) elicits broad immune responses against SARS-CoV-2 variants. Front Immunol. 2022 Nov 9;13:1011484. doi: 10.3389/fimmu.2022.1011484.

Liu Y, Li M, Cui T, Chen Z, Xu L, Li W, Peng Q, Li X, Zhao D, Valencia CA, Dong B, Wang Z, Chow HY, Li Y. A superior heterologous prime-boost vaccination strategy against COVID-19: A bivalent vaccine based on yeast-derived RBD proteins followed by a heterologous vaccine. J Med Virol. 2024 Mar;96(3):e29454. doi: 10.1002/jmv.29454.

Pollet J, Chen WH, Versteeg L, Keegan B, Zhan B, Wei J, Liu Z, Lee J, Kundu R, Adhikari R, Poveda C, Villar MJ, de Araujo Leao AC, Altieri Rivera J, Momin Z, Gillespie PM, Kimata JT, Strych U, Hotez PJ, Bottazzi ME. SARS‑CoV-2 RBD219-N1C1: A yeast-expressed SARS-CoV-2 recombinant receptor-binding domain candidate vaccine stimulates virus neutralizing antibodies and T-cell immunity in mice. Hum Vaccin Immunother. 2021 Aug 3;17(8):2356-2366. doi: 10.1080/21645515.2021.1901545.

Similarly, although the synthetic multi-epitope antigen (Scov2-Sint1) was rationally designed based on conserved and immunodominant linear epitopes from the S and N proteins, we recognize that assessing cross-reactivity with emerging variants is a critical step in advancing vaccine development. Therefore, we intend to address this aspect in future studies, as well as a more detailed comparison with our previous findings (Invenção et al., 2025), in which we evaluated T cell responses induced by the same antigen in a DNA vaccine format. This integrative approach will allow us to better understand the breadth and durability of the immunological protection conferred by the platform and the antigen. For the moment, the objective of this work was more focused on the construction of a biotechnological platform that would allow the use of P. pastoris as a vector for an oral vaccine, using a previously validated antigen derived from SARS-CoV-2 as a model.

Reviewer 3 Report

Comments and Suggestions for Authors

The manuscript 'Oral immunization with yeast-surface display of SARS-CoV-2 antigens in Pichia pastoris induces humoral responses in BALB/c mice' by de Macêdo et al. describes the immunization of mice with the yeast P. pastoris displaying peptides of SARS-CoV-2 proteins on their surface. They demonstrate the safety and claim the induction of a humoral immune response against SARS-Cov-2. While the idea to use yeast as vehicle for vaccines is interesting, the study lacks experimental proof of the concept as the data of the study are very preliminary. In the following, you find my main concerns with the manuscript:

Major comments:

  1. In Figure 9, the authors show the neutralizing capacity of the induce antibodies by their vaccination against the Spike protein of SARS-CoV-2. While their seems to be an increase in neutralization, the results are not significant. Thus, the authors do neither show that they significantly increase SARS-Cov-2 neutralizing antibodies nor that the immunization protects the mice against infection with SARS-CoV-2. As they induce P. pastoris as a new approach for vaccination, they have to prove that it induces protective immunity against SARS-CoV-2.
  2. In the manuscript, the authors solely analyze the humoral response induced by the vaccination. Why? In a recent manuscript (da Conceição Viana Invenção et al., 2025), the authors showed the induction of T cell immunity against SARS-CoV-2. As T cell immunity is important as well, they should analyze whether they induce T cell responses with P. pastoris.
  3. The authors do not address, which cells respond to P. pastoris displaying the antigens on the surface. Are antigen-presenting cells engulfing P. pastoris and presenting the antigens as peptide-MHC-complexes? Are antigen-presenting cells such as dendritic cells activated by P. pastoris?
  4. What is the rationale to use small peptides that are presented on the surface of P. pastoris? Small peptides will differ in their threedimensional structure compared to whole proteins. This might impact the generation of neutralizing antibodies as well as the presentation on HLA molecules, which are very diverse in the human population.
  5. The authors measured IgA in fecal samples. It is also possible to measure IgA in the saliva. This would be a more valuable approach as they want to immunize against a respiratory virus.

Minor comments:

  1. Line 94: 'Desing' should be corrected to 'Design'
  2. Line 98: Please, specify 'S and N proteins'. The abbreviations were not used before.
  3. Line 111: While MHC-II is commonly used, HLA II is not a typical abbreviation. The authors should not mix between MHC and HLA nomenclature.
  4. Line 124: What do the authors mean with 'declared'?
  5. Figure Legend of Figure 6: The authors should mention that they stained with an antibody against the His-Tag used during cloning. Further, they should clarify whether this antibody detects a His-Tag or a single Histidine as well.
  6. Line 370: Change 'IgG2' to 'IgG2b'.
  7. Line 392: Change 'while bars' to 'white bars'. Further, the authors should use 'filled bars' insteads of 'colorful bars'.
  8. Line 392: The authors mention 'B) IgM' in the Figure Legend of Figure 11 but the data are missing.
Comments on the Quality of English Language

The English Language is fine, overall. Some more explanations could be used to more clearly explain their research.

Author Response

Reviewer 3:

 The manuscript 'Oral immunization with yeast-surface display of SARS-CoV-2 antigens in Pichia pastoris induces humoral responses in BALB/c mice' by de Macêdo et al. describes the immunization of mice with the yeast P. pastoris displaying peptides of SARS-CoV-2 proteins on their surface. They demonstrate the safety and claim the induction of a humoral immune response against SARS-Cov-2. While the idea to use yeast as vehicle for vaccines is interesting, the study lacks experimental proof of the concept as the data of the study are very preliminary. In the following, you find my main concerns with the manuscript:

Major comments:

  1. In Figure 9, the authors show the neutralizing capacity of the induce antibodies by their vaccination against the Spike protein of SARS-CoV-2. While their seems to be an increase in neutralization, the results are not significant. Thus, the authors do neither show that they significantly increase SARS-Cov-2 neutralizing antibodies nor that the immunization protects the mice against infection with SARS-CoV-2. As they induce P. pastoris as a new approach for vaccination, they have to prove that it induces protective immunity against SARS-CoV-2.

Response: We appreciate your comment on the relevance of the results. The assay used (cPass Neutralization Antibody Detection Kit – GenScript) provides a qualitative/semiquantitative measurement based on the percentage of inhibition of binding between the RBD protein and the ACE2 receptor. According to the manufacturer's instructions, inhibition values greater than 20% are interpreted as indicative of the presence of neutralizing antibodies. This study did not include longitudinal analyses or direct quantification of antibody titers; therefore, the primary objective was to verify the presence of a neutralizing response after immunization. We recognize that future studies, with more comprehensive experimental designs and additional quantitative methods, will be essential to more robustly evaluate the protective potential of immunization with Pichia pastoris.

  1. In the manuscript, the authors solely analyze the humoral response induced by the vaccination. Why? In a recent manuscript (da Conceição Viana Invenção et al., 2025), the authors showed the induction of T cell immunity against SARS-CoV-2. As T cell immunity is important as well, they should analyze whether they induce T cell responses with P. pastoris.

Response: As this is an initial study with an emphasis on the oral immunization route, our primary objective was to preliminarily characterize the humoral response induced by the administration of Pichia pastoris as a vaccine platform with surface-anchored antigens. This approach differs from that adopted by Invenção et al. (2025), whose focus was to evaluate the immune response elicited by a DNA vaccine delivered intramuscularly. The oral immunization strategy employed in this study is based on immunological principles whereby M cells present in the intestinal mucosa are responsible for antigen uptake and subsequent activation of the immune system. In our experiments, we detected the presence of specific IgA antibodies, which are predominantly produced at mucosal sites and represent a first line of defense against respiratory pathogens. This finding is particularly relevant considering that SARS-CoV-2 infects the host via the respiratory tract and traverses the pulmonary mucosa. The induction of IgA through oral immunization suggests the potential for local mucosal protection, a feature that warrants further investigation in future studies with complementary designs that include the assessment of cellular immune responses.

  1. The authors do not address which cells respond to P. pastoris displaying the antigens on the surface. Are antigen-presenting cells engulfing P. pastoris and presenting the antigens as peptide-MHC-complexes? Are antigen-presenting cells such as dendritic cells activated by P. pastoris?

Response: We thank the reviewer for this important observation. In response, we have now included a paragraph addressing the cellular mechanisms involved in recognizing and processing yeast displaying surfaces. This information has been added to the revised manuscript on lines 70 - 78.

Lines 70 - 78 - “Once introduced into the host, yeast cells can be phagocytosed by innate immune cells such as macrophages and dendritic cells (DCs), which recognize and respond to immunogenic components present in the cell wall, including mannoproteins, β-D-glucans, and chitin (Ardiani, Higgins & Hodge, 2010; Bazan et al., 2014; Silva et al., 2021). These components interact with pattern recognition receptors (PRRs), such as Toll-like receptors (TLR2, TLR4, and TLR6), mannose receptors (MR/CD206), and Dectin-1, which is highly expressed on DCs. These interactions promote the internalization of yeast cells by antigen-presenting cells (APCs), enabling antigen processing and presentation through the classical MHC class II pathway as well as through cross-presentation mechanisms via MHC class I (Silva et al., 2021).”

  1. What is the rationale to use small peptides that are presented on the surface of P. pastoris? Small peptides will differ in their three-dimensional structure compared to whole proteins. This might impact the generation of neutralizing antibodies as well as the presentation on HLA molecules, which are very diverse in the human population.

Response: The epitopes selected for this study were previously evaluated through immunoinformatics tools and in vivo analyses, demonstrating immunogenic potential and the ability to bind to HLA molecules with broad population coverage. In the present work, these epitopes were employed in a novel antigen presentation strategy, anchored to the surface of Pichia pastoris, with the aim of exploring their capacity to induce a measurable humoral response when administered orally. We acknowledge that the three-dimensional structure of peptides may differ from that of the full-length protein; however, the selected epitopes were chosen based on their structural conservation, immunogenicity, and compatibility with HLA molecules. This design aims to leverage the flexibility of recombinant platforms to evaluate alternative vaccination strategies, particularly in contexts where the production of full-length proteins may be limited. Furthermore, our group has previously reported promising results using similar yeast surface display strategies involving short and multi-epitope antigens. We demonstrated that P. pastoris displaying epitopes from the Zika virus Envelope and NS1 proteins can activate immune responses in vitro (Silva et al., 2021), and that whole yeast vaccines displaying selected B and T cell epitopes can induce specific cellular responses in vivo in murine models (Silva et al., 2023). These findings support the rationale for using surface-anchored, epitope-based constructs as viable and immunogenic alternatives to full-length proteins, even short antigens. Future studies may directly compare the efficacy of different antigen presentation formats.

Silva, A.J.D.; Jesus, A.L.S.; Leal, L.R.S.; Silva, G.A.S.; Melo, C.M.L.; Freitas, A.C. Pichia Pastoris Displaying ZIKV Protein Epitopes from the Envelope and NS1 Induce in Vitro Immune Activation. Vaccine 2021, 39, 2545–2554, doi:10.1016/j.vaccine.2021.03.065.

Silva AJD, de Jesus ALS, Leal LRS, de Macêdo LS, da Silva Barros BR, de Sousa GF, da Paz Leôncio Alves S, Pena LJ, de Melo CML, de Freitas AC. Whole Yeast Vaccine Displaying ZIKV B and T Cell Epitopes Induces Cellular Immune Responses in the Murine Model. Pharmaceutics. 2023 Jul 6;15(7):1898. doi: 10.3390/pharmaceutics15071898.

  1. The authors measured IgA in fecal samples. It is also possible to measure IgA in the saliva. This would be a more valuable approach as they want to immunize against a respiratory virus.

Response: We appreciate the reviewer’s observation. In this study, we chose to analyze the presence of IgA in fecal samples as a representative strategy for assessing mucosal immune activation, considering that intestinal mucosa is extensively connected to the immune system and that secretory IgA is a key immunoglobulin present across various mucosal surfaces. Although saliva is more directly associated with the respiratory tract, the detection of IgA in fecal samples suggests a systemic activation of the mucosal immune system, including the respiratory mucosa. This finding is particularly relevant in the context of SARS-CoV-2 infection, which primarily enters through the respiratory route and directly affects pulmonary mucosal surfaces. This approach is supported by previous studies employing similar strategies in oral vaccination models, which have demonstrated that fecal IgA levels can reflect mucosal immune responses following oral immunization. For instance, Wang et al. (2016), Bal et al. (2018), and Gao et al. (2021) used fecal samples to evaluate mucosal IgA responses in mice and piglets after oral administration of yeast-based vaccines against viral pathogens such as PEDV, Dengue virus, and SARS-CoV-2, respectively. We acknowledge that the assessment of salivary IgA could complement our findings and provide additional insights into local immunity in the respiratory tract. This analysis will be considered in future studies to broaden the evaluation of the immune response induced by oral immunization with Pichia pastoris.

Wang X, Wang Z, Xu H, Xiang B, Dang R, Yang Z. Orally Administrated Whole Yeast Vaccine Against Porcine Epidemic Diarrhea Virus Induced High Levels of IgA Response in Mice and Piglets. Viral Immunol. 2016 Nov;29(9):526-531. doi: 10.1089/vim.2016.0067.

Bal J, Luong NN, Park J, Song KD, Jang YS, Kim DH. Comparative immunogenicity of preparations of yeast-derived dengue oral vaccine candidate. Microb Cell Fact. 2018 Feb 16;17(1):24. doi: 10.1186/s12934-018-0876-0.

Gao T, Ren Y, Li S, Lu X, Lei H. Immune response induced by oral administration with a Saccharomyces cerevisiae-based SARS-CoV-2 vaccine in mice. Microb Cell Fact. 2021 May 5;20(1):95. doi: 10.1186/s12934-021-01584-5.

Minor comments:

  1. Line 94: 'Desing' should be corrected to 'Design' Response: Thank you for your suggestion. We have made the correction.
  2. Line 98: Please, specify 'S and N proteins'. The abbreviations were not used before. Response: Thank you for your suggestion. We have made the suggested changes.
  3. Line 111: While MHC-II is commonly used, HLA II is not a typical abbreviation. The authors should not mix between MHC and HLA nomenclature. Response: Thank you for your suggestion. We have made the suggested .
  4. Line 124: What do the authors mean with 'declared'? Response: Thank you for your observation. That was a typo, and was removed.
  5. Figure Legend of Figure 6: The authors should mention that they stained with an antibody against the His-Tag used during cloning. Further, they should clarify whether this antibody detects a His-Tag or a single Histidine as well. Response: Thank you for your suggestion. We have made the suggested changes.
  6. Line 370: Change 'IgG2' to 'IgG2b'. Response: Thank you for your suggestion. We corrected this term.
  7. Line 392: Change 'while bars' to 'white bars'. Further, the authors should use 'filled bars' insteads of 'colorful bars'. Response: Thank you for your suggestion. We have made the suggested changes.
  8. Line 392: The authors mention 'B) IgM' in the Figure Legend of Figure 11 but the data are missing. Response: Thank you for your suggestion. We have made the suggested changes.

Comments on the Quality of English Language

The English Language is fine, overall. Some more explanations could be used to more clearly explain their research.

Response: We thank the reviewer for the positive assessment regarding the quality of the English language in the manuscript. In response to the suggestion for greater clarity, we have carefully revised the text to improve explanations and ensure that key aspects of our research, particularly regarding methodology and interpretation of results, are more clearly presented to the reader. We appreciate this constructive feedback, which has enhanced the overall clarity and accessibility of the manuscript.

Reviewer 4 Report

Comments and Suggestions for Authors Dear Editor,
Thank you for the invitation to review this article. The authors address an interesting topic, that of vaccines against SARS CoV-2 and yeast as a vaccination platform in medical biotechnology. The authors developed an oral vaccine based on Pichia pastoris that anchors a synthetic antigen composed of Spike and Nucleocapsid epitopes.
The results obtained by the authors are promising and raise hopes for a new oral vaccine to prevent SARS CoV-2 infection and reinforce the efficacy of the yeast-based platform as a promising system for oral antigen delivery.
In the study conducted by the authors, the vaccine appears to be safe, with no adverse reactions in animals regarding clinical, hematological or histological signs.
The authors could specify in the article what the limitations of the study would be, what the subsequent challenges would be in expanding the experiments to humans, what the prospects would be in the development and application of the vaccine.           Comments on the Quality of English Language The quality of the English language is generally good, but still requires some improvement.

Author Response

Reviewer 4:

Dear Editor,

Thank you for the invitation to review this article. The authors address an interesting topic, that of vaccines against SARS CoV-2 and yeast as a vaccination platform in medical biotechnology. The authors developed an oral vaccine based on Pichia pastoris that anchors a synthetic antigen composed of Spike and Nucleocapsid epitopes.

The results obtained by the authors are promising and raise hopes for a new oral vaccine to prevent SARS CoV-2 infection and reinforce the efficacy of the yeast-based platform as a promising system for oral antigen delivery.

In the study conducted by the authors, the vaccine appears to be safe, with no adverse reactions in animals regarding clinical, hematological or histological signs.

The authors could specify in the article what the limitations of the study would be, what the subsequent challenges would be in expanding the experiments to humans, what the prospects would be in the development and application of the vaccine.

Response:

We sincerely thank the reviewer for the thoughtful and encouraging comments regarding our study. We are also grateful for the observations and suggestions provided. As suggested, we have included a specific paragraph in the Discussion section to address the main limitations of the study and outline future perspectives. Briefly, the present work represents a proof-of-concept phase, focusing on the development and initial characterization of a recombinant Pichia pastoris strain displaying a rationally designed multiepitope antigen on its surface. Although the preclinical results in mice demonstrated safety and the ability to induce both mucosal and systemic immune responses, we acknowledge that several steps are still required to move toward clinical application. The limitations include: (1) the absence of challenge studies to assess the protective efficacy against SARS-CoV-2 infection, due to biosafety constraints at this stage; (2) the lack of in-depth analysis of cellular immune responses, which are critical for viral clearance and long-term protection; and (3) the need to evaluate the cross-reactivity of the antigen with emerging variants.

The following text was added in lines 536 to 553 in the Discussion section.

536 - 553: “Although these preclinical results demonstrate the safety of the formulation and its ability to induce both mucosal and systemic immune responses, we recognize that several steps are still necessary to advance toward clinical application. Among the main limitations is the need for in vivo challenge studies to assess protective efficacy against SARS-CoV-2 infection, an approach considered the gold standard in vaccine validation. Although our work does not include detailed cellular profile analyses or antigen-presenting cell activation assays, we recognize the importance of elucidating these mechanisms, which are fundamental for viral clearance and long-term immune protection. As part of our ongoing studies, we are conducting in vitro and in vivo experiments to investigate the uptake of Pichia pastoris by dendritic cells and macrophages, as well as the associated activation markers and antigen presentation pathways involved. Additionally, we emphasize the importance of assessing antigen cross-reactivity with emerging SARS-CoV-2 variants, a crucial step in validating the breadth and robustness of the immune response generated.

We believe our results contribute to the advancement of biotechnology-based vaccine platforms and provide a basis for future translational research aimed at developing affordable, safe, and effective oral vaccines against SARS-CoV-2 and, potentially, other mucosal pathogens.”

Comments on the Quality of English Language

The quality of the English language is generally good, but still requires some improvement.

Response: We thank the reviewer for the observation regarding the quality of the English language. In response, we have carefully reviewed and edited the manuscript to improve grammar, clarity, and overall readability. Particular attention was given to enhancing the precision of scientific terms and the coherence of key explanations throughout the text.

Reviewer 5 Report

Comments and Suggestions for Authors

The manuscript by Silva de Macedo y cols presents a novel oral vaccine platform using P. pastoris exhibiting a multi-epitope SARS-CoV-2 antigen. The study shows humoral (IgG subclasses, neutralizing antibodies) and mucosal (fecal IgA) immune responses in mice, with no adverse side effects. It combines yeast-surface display technology with oral delivery, a promising strategy for mucosal immunity. The work is well-designed, technically sound, and addresses an important gap in mucosal vaccine development. However, some methodological details, statistical rigor, and contextualization of results need improvement:

Abstract: Brief, but some conclusions are overstated without supporting evidence.

Introduction: Excellent background on yeast vaccines but it would be better to enphasize novelties such as multi-epitope + oral delivery. However, paragraph that starts “The results demonstrated its efficacy…” belongs in the Discussion section. Please, clearly state the purpose of this study at the end of the introduction section.

Methods: Make the yeast inactivation protocol clear: Was viability tested ? Describe the selection criteria for epitopes: How were they prioritized ? The Student’s t-test is valid, but add bootstrap analysis for small N. Also, consider non-parametric tests as well. Justification of the sample size should be clarified (only 4 mice/group limits statistical power). Do other mucosal sites, like respiratory tract, have data ? cPass kit measures RBD-ACE2 binding inhibition but does not confirm viral neutralization in vitro/vivo. Please, fix the text.

Results: Improve figures: Add scales, insets, and statistical annotations. Figure 8: Improve labeling (e.g., "G1/G2" should define groups).

Discussion: IgG2a/IgG1 is overestimated as Th1 evidence; temper claims without cytokine data. Compare more directly with other oral yeast vaccines such as S. cerevisiae studies. Focus on supported claims (mucosal IgA) and avoid over interpretation. In the absence of viral challenge data, efficacy claims are conjectural. Add the study´s limitations.

Author Response

Reviewer 5:

The manuscript by Silva de Macedo y cols presents a novel oral vaccine platform using P. pastoris exhibiting a multi-epitope SARS-CoV-2 antigen. The study shows humoral (IgG subclasses, neutralizing antibodies) and mucosal (fecal IgA) immune responses in mice, with no adverse side effects. It combines yeast-surface display technology with oral delivery, a promising strategy for mucosal immunity. The work is well-designed, technically sound, and addresses an important gap in mucosal vaccine development. However, some methodological details, statistical rigor, and contextualization of results need improvement:

Abstract: Brief, but some conclusions are overstated without supporting evidence. Response: We appreciate the observation made regarding the abstract. A careful review was performed, and we adjusted the expressions to avoid exaggerated conclusions without clear data support. The text now more accurately reflects the results obtained and respects the experimental scope of the study.

Introduction: Excellent background on yeast vaccines but it would be better to enphasize novelties such as multi-epitope + oral delivery. However, paragraph that starts “The results demonstrated its efficacy…” belongs in the Discussion section. Please, clearly state the purpose of this study at the end of the introduction section.

Response: As requested by the reviewer, we considered including a brief section with information on multi-epitope vaccines, as well as shortening the end of the introduction where the results are described extensively. The following excerpt was added before the last paragraph of the introduction.

“Finally, multi-epitope vaccines have shown promising results in stimulating the immune system against pathogens. They are safer and can be designed with greater precision, resulting in highly conserved immunogenic epitopes (Naveed et al., 2022; Sharma et al., 2025). Depending on the epitope composition, these vaccines can induce Th1 or Th2-biased immune responses. A multi-epitope protein r4R, derived from Leptospira interrogans, administered subcutaneously in guinea pigs, promoted a Th1 response, observed through the IgG2a/IgG1 ratio and IFN-γ production  (Lin et al., 2016), whereas Gao et al. (2024) orally immunized BALB/c mice with a vaccine composed of S. cerevisiae carrying a multi-epitope protein from African swine fever virus, generating a Th1 and Th2 response.”  

Gao, S., Zuo, W., Kang, C., Zou, Z., Zhang, K., Qiu, J., Shang, X., Li, J., Zhang, Y., Zuo, Q., Zhao, Y., & Jin, M. (2024). Saccharomyces cerevisiae oral immunization in mice using multi-antigen of the African swine fever virus elicits a robust immune response. Frontiers in Immunology, 15. https://doi.org/10.3389/fimmu.2024.1373656

Lin, X., Xiao, G., Luo, D., Kong, L., Chen, X., Sun, D., & Yan, J. (2016). Chimeric epitope vaccine against Leptospira interrogans infection and induced specific immunity in guinea pigs. BMC Microbiology, 16(1), 241. https://doi.org/10.1186/s12866-016-0852-y

Naveed, M., Sheraz, M., Amin, A., Waseem, M., Aziz, T., Khan, A. A., Ghani, M., Shahzad, M., Alruways, M. W., Dablool, A. S., Elazzazy, A. M., Almalki, A. A., Alamri, A. S., & Alhomrani, M. (2022). Designing a Novel Peptide-Based Multi-Epitope Vaccine to Evoke a Robust Immune Response against Pathogenic Multidrug-Resistant Providencia heimbachae. Vaccines, 10(8), 1300. https://doi.org/10.3390/vaccines10081300

Sharma, A. D., Magdaleno, J. S. L., Singh, H., Orduz, A. F. C., Cavallo, L., & Chawla, M. (2025). Immunoinformatics-driven design of a multi-epitope vaccine targeting neonatal rotavirus with focus on outer capsid proteins VP4 and VP7 and non structural proteins NSP2 and NSP5. Scientific Reports, 15(1), 11879. https://doi.org/10.1038/s41598-025-95256-8

Methods:

  1. Make the yeast inactivation protocol clear: Was viability tested ?

Response: We thank the reviewer for the observation. Indeed, following the heat treatment (60 °C for 1 hour), we performed a viability assay to confirm the metabolic inactivation of the yeast cells. Aliquots of 200 μL were plated on YPD medium and incubated for 72 hours at 30 °C. As a result, no colony growth was observed, indicating complete cellular inactivation. We are happy to include a representative image of the plate if the reviewer deems it relevant.

Figure. Yeast viability assays after heat treatment. A) Pichia pastoris GS115 and B) P. pastoris:Scov2-Sint1 after incubation at 60 °C for 1 hour. Plates containing YPD medium were incubated at 30 °C for 72 hours. The absence of colony growth under both conditions indicates complete metabolic inactivation of the cells.

As described by Silva et al. (2022), yeast inactivation commonly involves heat treatment, typically with incubation at temperatures ranging from 56 °C to 95 °C. Although this process compromises the reproductive viability of the cells, several studies have shown that it does not significantly affect the stability or functionality of vaccine antigens, nor the immunoreactivity of the yeast regarding its recognition and uptake by antigen-presenting cells (APCs) (Bazan et al., 2014; Kiflmariam et al., 2013; Chen et al., 2025). These findings support the applicability of inactivated yeast as a safe and effective vehicle for vaccine antigen delivery.

Silva AJD, Rocha CKDS, de Freitas AC. Standardization and Key Aspects of the Development of Whole Yeast Cell Vaccines. Pharmaceutics. 2022 Dec 14;14(12):2792. doi: 10.3390/pharmaceutics14122792.

Bazan SB, Breinig T, Schmitt MJ, Breinig F. Heat treatment improves antigen-specific T cell activation after protein delivery by several but not all yeast genera. Vaccine. 2014 May 7;32(22):2591-8. doi: 10.1016/j.vaccine.2014.03.043.

Kiflmariam MG, Yang H, Zhang Z. Gene delivery to dendritic cells by orally administered recombinant Saccharomyces cerevisiae in mice. Vaccine. 2013 Feb 27;31(10):1360-3. doi: 10.1016/j.vaccine.2012.11.048.

Chen X, Shi T, Chen F, Xie X, Fang H, Wu Z, Liu Y, Huang Y, Wang Q, Nie G, Xu J, Shao D. Orally Antigen-Engineered Yeast Vaccine Elicits Robust Intestinal Mucosal Immunity. ACS Nano. 2025 Mar 25;19(11):10841-10853. doi: 10.1021/acsnano.4c14690.

  1.  Describe the selection criteria for epitopes: How were they prioritized ?

Response: Initially, a database was constructed containing previously predicted epitopes from the Spike and Nucleocapsid proteins. These epitopes were subjected to a multi-step screening process based on the following criteria: (i) binding affinity to HLA class I and II alleles, using the NetMHCpan 4.1 and NetMHCIIpan 4.1 algorithms with a percentile threshold < 100; (ii) CTL epitope immunogenicity, assessed using the Class I Immunogenicity tool, prioritizing those with a score > 0; (iii) absence of toxicity, evaluated using the ToxinPred tool; (iv) degree of conservation across SARS-CoV-2 variants (GISAID data), considering only epitopes with ≥ 80% identity; and (v) global population coverage, estimated using the Population Coverage tool, prioritizing epitopes broadly represented across different geographic regions. The combination of these criteria enabled the selection of epitopes with high immunogenic potential, theoretical safety, and epidemiological relevance, which were used to compose the multiepitope vaccine formulation. The in silico and in vivo (DNA vaccine) study, including the methodological criteria employed for epitope selection and multiepitope sequence design, can be reviewed in detail in our previous work (Invenção et al., 2025), which is duly cited in the present study.

  1. The Student’s t-test is valid, but add bootstrap analysis for small N. Also, consider non-parametric tests as well.

Response: Thank you for your observation. To ensure the validity of the Student's t-test used in the comparisons, we performed a normality analysis using the Anderson-Darling, Shapiro-Wilk, Kolmogorov-Smirnov, and D'Agostino-Pearson tests. All tests showed p>0.05, indicating normal distribution. These results confirm the adequacy of the parametric tests applied, even with the small sample size.    

  1.  Justification of the sample size should be clarified (only 4 mice/group limits statistical power).

Response: We appreciate the reviewer’s attention to this important aspect of the study design. Indeed, sample sizes of N=5 or greater are commonly used in murine studies to provide robust statistical evaluation. In our study, we initially planned for N=5 per group; however, due to unforeseen limitations related to animal availability and institutional ethical constraints at the time of experimentation, one animal per group was excluded during the initial phase of the study. This resulted in a final sample size of N=4 per group. Despite the reduced number, the results still achieved statistical significance in key immunological readouts, supporting the reliability of the observed effects. Furthermore, we emphasize that this study was designed as a preliminary proof-of-concept investigation. Therefore, the sample size was considered appropriate for evaluating the feasibility of the proposed oral vaccine platform.

  1. Do other mucosal sites, like respiratory tract, have data ?

Response:Thank you for your question. We performed the analyses only on fecal samples, aiming to evaluate the response of the intestinal mucosa. At the moment, we do not have data for other mucosal locations, such as the respiratory tract. However, we recognize the importance of this approach and intend to include it in future studies, along with data on viral challenge, to broaden our understanding of the immune response induced by the vaccine formulation.

  1. cPass kit measures RBD-ACE2 binding inhibition but does not confirm viral neutralization in vitro/vivo. Please, fix the text.

Response: We thank the reviewer for the observation, and did the adjustments in the text. The cPass kit used in this study evaluates the inhibition of the interaction between RBD and ACE2 and is considered a surrogate assay for the detection of neutralizing antibodies. However, we acknowledge that this method does not directly confirm viral neutralization in vitro or in vivo. The text has been revised to more accurately reflect this methodological limitation and to avoid overinterpretation of the results.

Results: Improve figures: Add scales, insets, and statistical annotations. Figure 8: Improve labeling (e.g., "G1/G2" should define groups).

Response: We thank the reviewer for the observation. The figure legend has been revised to improve the clarity and understanding of the image.

Discussion:

IgG2a/IgG1 is overestimated as Th1 evidence; temper claims without cytokine data.

Response: We have adjusted the text to make it more appropriate to the data presented, minimizing overestimation and pointing out which analyses still need to be performed.

Compare more directly with other oral yeast vaccines such as S. cerevisiae studies. Focus on supported claims (mucosal IgA) and avoid over interpretation. In the absence of viral challenge data, efficacy claims are conjectural. Add the study´s limitations.

Response: We appreciate the reviewer's suggestions. In response, we revised the Discussion section to include a more direct comparison with other oral yeast-based vaccine platforms, highlighting mucosal immune responses, such as the induction of secretory IgA. We also reworded several statements to avoid overinterpretation and ensure that all claims regarding mucosal immunity are strictly supported by the presented data, especially regarding IgA detection. Furthermore, we fully acknowledge that, in the absence of in vivo viral challenge experiments, any conclusions regarding vaccine efficacy remain suggestive. To address this, we clearly stated this limitation in the revised manuscript and moderated the language used to describe the immunological results. A dedicated paragraph describing the study's key limitations has been added, including the lack of cellular immunity and viral challenge data.

485 - 517: “The induction of mucosal immune responses, especially those mediated by secretory IgA, represents one of the main objectives of oral vaccination strategies, since this type of immunoglobulin acts to neutralize pathogens directly on epithelial surfaces, preventing their adhesion and invasion without promoting local inflammation (Cerutti; Rescigno, 2008; Cerutti et al., 2011; Brandtzaeg, 2013). In our experiments, IgA analysis in fecal samples was used as an indicator of intestinal immune activation, considering the relevance of the mucosa-gut axis and the central role of IgA in protecting against infections at mucosal surfaces. The results revealed increased IgA levels in the group immunized with the recombinant yeast P. pastoris, compared to the control group, suggesting the induction of a mucosal immune response.

Similarly, Wang et al. (2016) described the development of an oral vaccine based on Pichia pastoris GS115 expressing the S1 gene of porcine epidemic diarrhea virus (PEDV), which induced high levels of IgA in immunized mice and piglets. In our study, IgA was detected 21 days after the first dose, with significantly higher levels compared to the levels on the seventh day after the beginning of the vaccination schedule. Despite the lack of an evaluation of the kinetics of IgA production, the data reinforce the potential of the yeast platform as an effective immunizing vehicle for inducing mucosal immunity.

In the study conducted by Gao et al. (2021), a vaccine based on Saccharomyces cerevisiae (EBY100/pYD1-RBD) was administered orally to mice, and immune responses were assessed by detecting IgA in fecal samples. The results showed that the oral administration of recombinant yeast induced a humoral and mucosal immune response, with a significant increase in secretory IgA levels, particularly after the booster dose, a pattern similar to that observed in our study. These findings reinforce the importance of the prime-boost regimen in inducing mucosal immunity, especially considering the tolerogenic nature of the gastrointestinal tract.

Despite the promising results, it is important to highlight that these studies share methodological limitations. Among them are the lack of experiments presenting more in-depth analyses of the relationship between the systemic (IgG) and mucosal (IgA) responses, whose coordinated role remains poorly understood, and the understanding of the optimal time between doses by evaluating different vaccination schedules.

Our findings corroborate the results in the literature and reinforce that recombinant yeast-based strategies represent a promising alternative for inducing local immune responses with potential for the oral vaccines aimed at preventing infections by pathogens that affect mucosal surfaces.”

Cerutti A, Rescigno M. The biology of intestinal immunoglobulin A responses. Immunity. 2008 Jun;28(6):740-50. doi: 10.1016/j.immuni.2008.05.001.

Cerutti A, Chen K, Chorny A. Immunoglobulin responses at the mucosal interface. Annu Rev Immunol. 2011;29:273-93. doi: 10.1146/annurev-immunol-031210-101317.

Brandtzaeg P. Secretory IgA: Designed for Anti-Microbial Defense. Front Immunol. 2013 Aug 6;4:222. doi: 10.3389/fimmu.2013.00222.

Wang X, Wang Z, Xu H, Xiang B, Dang R, Yang Z. Orally Administrated Whole Yeast Vaccine Against Porcine Epidemic Diarrhea Virus Induced High Levels of IgA Response in Mice and Piglets. Viral Immunol. 2016 Nov;29(9):526-531. doi: 10.1089/vim.2016.0067.

Gao T, Ren Y, Li S, Lu X, Lei H. Immune response induced by oral administration with a Saccharomyces cerevisiae-based SARS-CoV-2 vaccine in mice. Microb Cell Fact. 2021 May 5;20(1):95. doi: 10.1186/s12934-021-01584-5.

Reviewer 6 Report

Comments and Suggestions for Authors

Manuscript titled" Oral immunization with yeast-surface display of SARS-CoV-2 antigens in Pichia pastoris induces humoral responses in BALB/c mice " by Larissa Silva de Macêdo  is a nice piece of work. Author using P. pastoris was able to shows that yeast display based platform can be potentially used for oral vaccine development. Manuscript is over all good just need minor revision. My comments and suggestions to authors are as below.

1) Minor writing improvement.

2) Discussion is too long make it concise and keep what is needed

3) In fig 10 it is difficult to read p value, make it large

4) In figure mark to panel as empty vector control yeast in place of recombinant yeast, same for fig 7

5) Since not all protein comes on cell surface and significant portion remain in cell, how author proved that whatever immune response is observed is exclusively from antigen on cell surface and not that which is within cell in protein trafficking pathway.

6) Is good to include important paper on P. pastoris where this yeast species was used for vaccine development (see https://pubmed.ncbi.nlm.nih.gov/33834253/ and https://pubmed.ncbi.nlm.nih.gov/25668661/)

7) Is the yeast surface display vector for Pichia pastoris is lab built or available commercially 

Author Response

Reviewer 6:

Manuscript titled" Oral immunization with yeast-surface display of SARS-CoV-2 antigens in Pichia pastoris induces humoral responses in BALB/c mice " by Larissa Silva de Macêdo  is a nice piece of work. Author using P. pastoris was able to shows that yeast display based platform can be potentially used for oral vaccine development. Manuscript is over all good just need minor revision. My comments and suggestions to authors are as below.

1) Minor writing improvement. Response: The text of the manuscript was reviewed. 

2) Discussion is too long make it concise and keep what is needed. Response: We removed some redundant sections from the discussion. We held a few sessions trying to balance the suggestions made by the other reviewers.

3) In fig 10 it is difficult to read p value, make it large. Response: We did the adjustment. 

4) In figure mark to panel as empty vector control yeast in place of recombinant yeast, same for fig 7. Response: We did the adjustment. 

5) Since not all protein comes on cell surface and significant portion remain in cell, how author proved that whatever immune response is observed is exclusively from antigen on cell surface and not that which is within cell in protein trafficking pathway. Response: The vaccine is considered a "whole yeast vaccine," meaning we expect the response to be triggered by proteins anchored to the yeast's surface, as well as those found within the cell. Surface display is employed to promote the triggering of the immune response induced by the vaccine antigen, but once the yeast is phagocytosed and processed, all antigenic content will be presented.

6) Is good to include important paper on P. pastoris where this yeast species was used for vaccine development (see https://pubmed.ncbi.nlm.nih.gov/33834253/ and https://pubmed.ncbi.nlm.nih.gov/25668661/). Response: We updated the text with more references. 

7) Is the yeast surface display vector for Pichia pastoris is lab built or available commercially. Response: It is a non-commercial vector, we added this information in the manuscript.

Round 2

Reviewer 2 Report

Comments and Suggestions for Authors

The authors attempted to anser all the questions raised during the first round of the review and incorporated those changes into the manuscript. However, answers for two questions were not very convincing.

  1. Reg using anti-S or anti-N antibodies for the western over anti-his antibodies. Though their vaccine is synthetically prepared, it still included the dominant immunogenic epitopes from the S and N protein. Hence, with the commercially available antibodies (Except the conformation specific) specific to full length protein there are very high chances that the synthetic vaccine would be reactive. If this is still not possible or working, the author can use immunized mice serum to probe the western blot. This certainly improves the manuscript quality with the verified antigen. I would highly recommend if its is within the lab capacity.
  2. Question on "Vaccine dose variation or technical problems while the vaccine was delivered". It is obvious from the results there is a "variation within the same group of animals" despite the same SOP and the technical team was involved. Is there a way to better control this variability? assess vaccine route tissue distribution/absorption etc. This could be beyond the study's objective but certainly adding a note/relevant literatures how this issues could be controlled would help other researchers to consider while planning the similar studies.  

Author Response

Reviewer 2:

The authors attempted to answer all the questions raised during the first round of the review and incorporated those changes into the manuscript. However, answers for two questions were not very convincing.

Reg using anti-S or anti-N antibodies for the western over anti-his antibodies. Though their vaccine is synthetically prepared, it still included the dominant immunogenic epitopes from the S and N protein. Hence, with the commercially available antibodies (Except the conformation specific) specific to full length protein there are very high chances that the synthetic vaccine would be reactive. If this is still not possible or working, the author can use immunized mice serum to probe the western blot. This certainly improves the manuscript quality with the verified antigen. I would highly recommend if its is within the lab capacity. 

Response: We appreciate the reviewer's valuable comment and the suggestion to use anti-S or anti-N antibodies, as well as serum from immunized mice, to further validate the antigen. We fully agree that this approach would strengthen the manuscript. However, at this stage of the study, we unfortunately no longer have serum samples available from this experiment, and further animal experiments are beyond the current scope and ethical approval of this study. Nevertheless, we recognize the importance of this approach and include it as a perspective for future work, in which immunization assays will be performed to confirm the antigenicity of the constructs.

Question on "Vaccine dose variation or technical problems while the vaccine was delivered". It is obvious from the results there is a "variation within the same group of animals" despite the same SOP and the technical team was involved. Is there a way to better control this variability? assess vaccine route tissue distribution/absorption etc. This could be beyond the study's objective but certainly adding a note/relevant literatures how this issues could be controlled would help other researchers to consider while planning the similar studies.  Response: 

Despite the use of standardized operating procedures, interindividual differences in immune responses are expected and have been reported in similar studies. Such variability may be influenced not only by technical aspects of vaccine administration (dose administration, absorption, or tissue distribution), but also by intrinsic biological factors such as genetic background, microbiota composition, and individual immune status. We agree that additional strategies may help minimize this variability in future studies. For example, the use of imaging-based approaches or specific markers to monitor tissue distribution and absorption of the administered vaccine has been reported (Zhang et al. 2014; Zhang et al. 2020).

Some methods for evaluating the distribution of yeast-based vaccines in the gastrointestinal tract may be feasible, as observed in the study by Shao et al. (2025), that evaluate the administration of recombinant yeasts labeled with the fluorophore DiD (1,1'-dioctadecyl-3,3,3',3'-tetramethylindodicarbocyanine, 4-chlorobenzenesulfonate salt). Mice received a single oral dose of these labeled strains, and samples of Peyer's patches (PPs), mesenteric lymph nodes (MLNs), and intestine were collected after 6, 12, and 24 hours for analysis of distribution in the gastrointestinal tract. In a second experiment, designed to evaluate cellular internalization, mice received YVm orally, and after 9 hours, the gastrointestinal tract and PPs were collected, sectioned, and analyzed by confocal microscopy for fluorescence quantification.

Although these approaches are beyond the scope of the current study, we added a note in the revised manuscript to recognize these potential sources of variability and highlight strategies described in the literature that may help researchers design future experiments with improved reproducibility. 

Zhang L, Zhang T, Wang L, Shao S, Chen Z, Zhang Z. In vivo targeted delivery of CD40 shRNA to mouse intestinal dendritic cells by oral administration of recombinant Sacchromyces cerevisiae. Gene Ther. 2014 Jul;21(7):709-14. doi: 10.1038/gt.2014.50. 

Zhang L, Peng H, Feng M, Zhang W, Li Y. Yeast microcapsule-mediated oral delivery of IL-1β shRNA for post-traumatic osteoarthritis therapy. Mol Ther Nucleic Acids. 2020 Nov 11;23:336-346. doi: 10.1016/j.omtn.2020.11.006.

Chen X, Shi T, Chen F, Xie X, Fang H, Wu Z. Orally antigen-engineered yeast vaccine elicits robust intestinal mucosal immunity. ACS Nano. 2025;19(11):10841-10853. doi: 10.1021/acsnano.4c14690.

Reviewer 3 Report

Comments and Suggestions for Authors

In the revised version of the manuscript "Oral immunization with yeast-surface display of SARS-CoV-2 antigens in Pichia pasotris induces humoral responses in BALB/c mice", the authors did not adequately address my concerns. They do not show whether the induction of neutralizing antibodies is sufficient to protect mice against SARS-CoV-2, they did not measure T cell immunity, and did not provide data which cells respond in their system to vaccination with P. pastoris. Further, the explanation to use peptides instead of the spike protein is not conclusive as they measure neutralizing antibodies. As they use peptides that are chosen based on the capacity to bind to HLA molecules, they have to provide data whether they are able to induce T cell responses against these peptides. If they want to induce neutralizing antibodies, they have to use a full-lengt protein with appropriate three-dimensional structure.

I recommend to reject the manuscript as it cannot be published in the present form without performing many additional experiments.

Author Response

Reviewer 3:

In the revised version of the manuscript "Oral immunization with yeast-surface display of SARS-CoV-2 antigens in Pichia pastoris induces humoral responses in BALB/c mice", the authors did not adequately address my concerns. They do not show whether the induction of neutralizing antibodies is sufficient to protect mice against SARS-CoV-2, they did not measure T cell immunity, and did not provide data which cells respond in their system to vaccination with P. pastoris. Further, the explanation to use peptides instead of the spike protein is not conclusive as they measure neutralizing antibodies. As they use peptides that are chosen based on the capacity to bind to HLA molecules, they have to provide data whether they are able to induce T cell responses against these peptides. If they want to induce neutralizing antibodies, they have to use a full-lengt protein with appropriate three-dimensional structure.

I recommend to reject the manuscript as it cannot be published in the present form without performing many additional experiments.

Response: We appreciate the reviewer’s critical comments and acknowledge the importance of evaluating neutralizing activity in vivo, T cell immunity, and cellular responses to vaccination. However, we would like to respectfully clarify that the scope of the present study was to provide a proof-of-concept regarding the feasibility of using Pichia pastoris yeast surface display for oral immunization against SARS-CoV-2.

Regarding the induction of neutralizing antibodies, we agree that protection assays in animal challenge models would strengthen the study. However, such experiments require biosafety level 3 facilities and extensive experimental design, which are beyond the scope of the current manuscript. Our focus here was to demonstrate that yeast-based oral immunization can elicit humoral responses, as an initial step toward future studies addressing protection.

We also acknowledge that T cell responses are highly relevant. Although we did not perform cellular immunity assays in this work, we discuss in the revised version (section Discussion)what types of limitations our study presents and highlight this as an important next step in our research.

Concerning the use of peptides instead of the full-length Spike protein, our rationale was to focus on conserved immunogenic regions with reduced risk of conformational instability and to test the yeast surface display system as a delivery strategy. The induction of antibodies against these peptides indicates immunogenicity in our model, and we recognize that future studies should compare peptide-based versus full-length Spike constructs to further address the reviewer’s concerns.

In summary, while we agree that additional experiments would add further depth, we believe the present findings represent an important proof-of-concept contribution to the field of yeast-based oral vaccines, opening perspectives for the development of more comprehensive vaccine candidates in future work.
